# Diverse mutational landscapes in human lymphocytes

Heather E. Machado[1], Emily Mitchell[1,2,17], Nina F. Øbro[2,3,4,17], Kirsten Kübler[5,6,7,17], Megan Davies[2,3,8], Daniel Leongamornlert[1], Alyssa Cull[9], Francesco Maura[10], Mathijs A. Sanders[1,11], Alex T. J. Cagan[1], Craig McDonald[2,3,9], Miriam Belmonte[2,3,9], Mairi S. Shepherd[2,3], Felipe A. Vieira Braga[1], Robert J. Osborne[1,12], Krishnaa Mahbubani[3,13,14], Iñigo Martincorena[1], Elisa Laurenti[2,3], Anthony R. Green[2,3], Gad Getz[5,6,7,15], Paz Polak[16], Kourosh Saeb-Parsy[13,14], Daniel J. Hodson[2,3], David G. Kent[2,3,9] ✉ & Peter J. Campbell[1,2] ✉

The lymphocyte genome is prone to many threats, including programmed mutation during differentiation[1], antigen-driven proliferation and residency in diverse microenvironments. Here, after developing protocols for expansion of single-cell lymphocyte cultures, we sequenced whole genomes from 717 normal naive and memory B and T cells and haematopoietic stem cells. All lymphocyte subsets carried more point mutations and structural variants than haematopoietic stem cells, with higher burdens in memory cells than in naive cells, and with T cells accumulating mutations at a higher rate throughout life. Off-target effects of immunological diversification accounted for approximately half of the additional differentiation-associated mutations in lymphocytes. Memory B cells acquired, on average, 18 off-target mutations genome-wide for every on-target *IGHV* mutation during the germinal centre reaction. Structural variation was 16-fold higher in lymphocytes than in stem cells, with around 15% of deletions being attributable to off-target recombinase-activating gene activity. DNA damage from ultraviolet light exposure and other sporadic mutational processes generated hundreds to thousands of mutations in some memory cells. The mutation burden and signatures of normal B cells were broadly similar to those seen in many B-cell cancers, suggesting that malignant transformation of lymphocytes arises from the same mutational processes that are active across normal ontogeny. The mutational landscape of normal lymphocytes chronicles the off-target effects of programmed genome engineering during immunological diversification and the consequences of differentiation, proliferation and residency in diverse microenvironments.

The adaptive immune system depends on programmed somatic mutation to generate antigen receptor diversity. T cells use recombinase-activating gene (RAG)-mediated deletion to generate functional T cell receptors (TCRs); B cells also use RAG-mediated deletion to rearrange immunoglobulin (Ig) heavy and light chains, followed by activation-induced cytidine deaminase (AID)-mediated somatic hypermutation (SHM) and class-switch recombination (CSR) to further increase diversity[1]. Off-target genome editing in lymphocytes can produce mutations driving lymphoid malignancies, including RAG-mediated deletions in acute lymphoblastic leukaemia[2,3];

AID-mediated SHM in diffuse large B cell lymphoma[4–6]; and CSR in multiple myeloma[7].

Although the accumulation of mutations in lymphoid malignancies is well characterized, the mutation burden of normal lymphocytes has been less comprehensively studied. Patterns of base substitutions in 59 normal, CD19-positive B cells revealed an age-related increase in burden, with evidence for off-target SHM[8]. More detailed quantification and comparison of the genomic landscape of B versus T cells, naive versus memory cells, and normal versus malignant lymphocytes is lacking.

[1]Wellcome Sanger Institute, Hinxton, UK. [2]Wellcome MRC Cambridge Stem Cell Institute, University of Cambridge, Cambridge, UK. [3]Department of Haematology, University of Cambridge, Cambridge, UK. [4]Department of Clinical Immunology, Copenhagen University Hospital, Rigshospitalet, Copenhagen, Denmark. [5]Broad Institute of MIT and Harvard, Cambridge, MA, USA. [6]Center for Cancer Research, Massachusetts General Hospital, Charlestown, MA, USA. [7]Harvard Medical School, Boston, MA, USA. [8]Cambridge Molecular Diagnostics, Milton Road, Cambridge, United Kingdom. [9]York Biomedical Research Institute, University of York, Wentworth Way, York, United Kingdom. [10]Sylvester Comprehensive Cancer Center, Miami, Florida, USA. [11]Department of Hematology, Erasmus MC Cancer Institute, Rotterdam, The Netherlands. [12]Biofidelity, 330 Cambridge Science Park, Milton Road, Cambridge, United Kingdom. [13]Department of Surgery, University of Cambridge, Cambridge, United Kingdom. [14]NIHR Cambridge Biomedical Research Centre, Cambridge Biomedical Campus, Cambridge, United Kingdom. [15]Department of Pathology, Massachusetts General Hospital, Boston, Massachusetts, USA. [16]Oncological Sciences, Icahn School of Medicine at Mount Sinai, New York, USA. [17]These authors contributed equally: Emily Mitchell, Nina F Øbro, Kirsten Kübler. ✉e-mail: david.kent@york.ac.uk; pc8@sanger.ac.uk

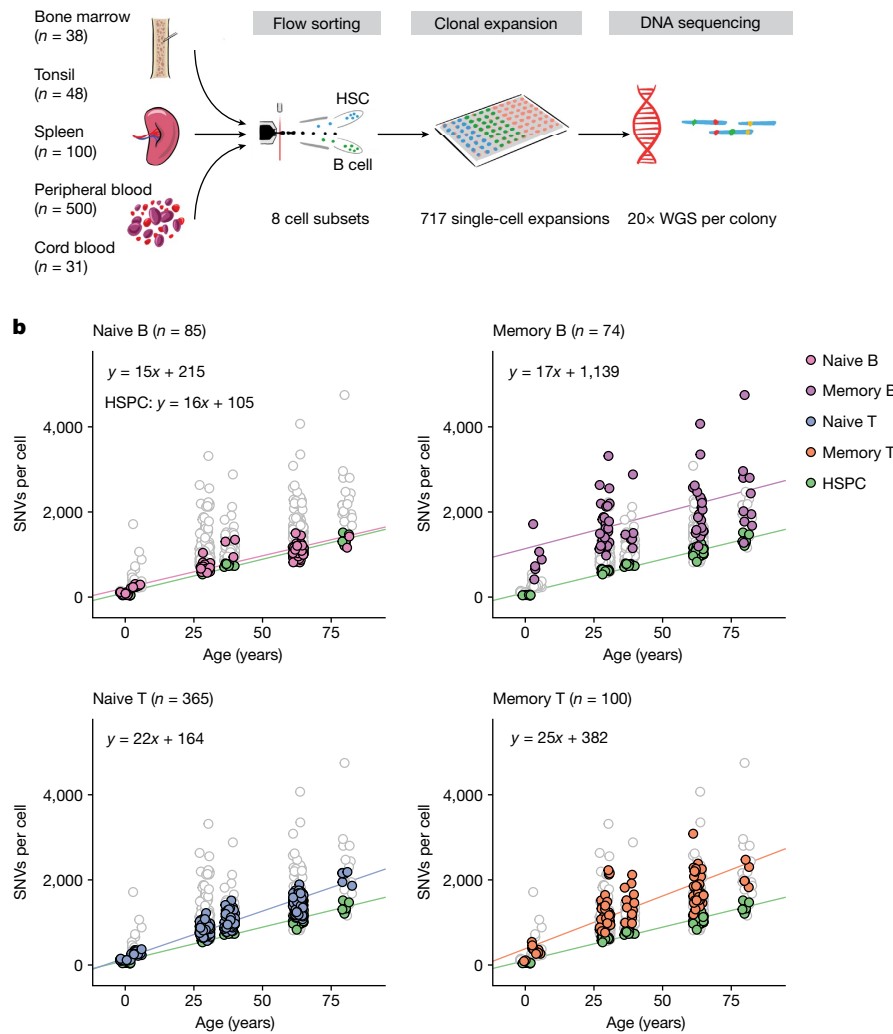

**Fig. 1 | Experimental design and lymphocyte mutation burden with age.**
**a**, Schematic of the experimental design. WGS, whole-genome sequencing.
**b**, SNV mutation burden per genome for the four main lymphocyte subsets, compared with HSPCs (green points). Each panel shows data for HSPCs and the indicated cell type in colour, with the other three lymphocyte subsets plotted in white with grey outline. The lines show the fit for the indicated cell type using linear mixed-effects models.

## Genome sequencing of B and T cells

Expanding single cells into colonies in vitro enables accurate identification of all classes of somatic mutation using genome sequencing[9–11]. We developed protocols for expanding flow-sorted single naive and memory B and T cells in vitro to colonies of 30 to more than 2,000 cells (Fig. 1a, Supplementary Fig. 1 and Methods). Culture efficiencies varied by cell type, but were typically 2–5% (Supplementary Table 1), which prompted us to evaluate whether there was evidence for potential bias in culture efficiency among lymphocytes (Supplementary Note). Reassuringly, cell surface marker expression was similar between lymphocytes that grew into colonies and those that did not (Extended Data Fig. 1). Furthermore, deep sequencing data for one donor showed strong correlation between variant allele fractions in bulk lymphocytes versus colonies (Extended Data Fig. 2a)—using bootstrapping, we estimate that any bias in culture efficiency among lineages would amount to just 20% (for example, ranging from 0.04–0.06 for a mean efficiency of 0.05) for both B and T cells (Supplementary Note).

We obtained blood, spleen and bone marrow samples from four individuals aged 27–81 years, as well as tonsillar tissue from two 4-year-old children and cord blood from a neonate (Supplementary Table 2). All individuals studied were haematopoietically normal and healthy; one had a history of inflammatory bowel disease treated with azathioprine and the two tonsil donors had a history of tonsillitis. We focused on four classes of lymphocytes: naive B cells, memory B cells, CD4+ and CD8+ naive T cells, and CD4+ and CD8+ memory T cells. We also expanded T regulatory cells from one subject. Five of the subjects reported here were also analysed in a parallel study[12] of haematopoietic stem and progenitor cells (HSPCs) with 39 overlapping HSPC genomes.

We performed whole-genome sequencing to an average depth of approximately 20×. To confirm that this provided sufficient depth, we calculated recall statistics for germline heterozygous variants for each colony, generating estimates of sensitivity of 80% at 10× and more than 98% at 20× depth (Extended Data Fig. 2b). The final dataset comprises 717 whole genomes (Supplementary Table 3).

## Mutation burden

The overall burden of both single nucleotide variants (SNVs) and insertion–deletions (indels) per cell varied extensively, influenced predominantly by age and cell type (Fig. 1b). The burden of SNVs increased linearly with age across all cell types, but the rate of mutation accumulation differed across cell types ($P = 1 \times 10^{-4}$ for the age–cell type interaction; linear mixed-effects model). HSPCs accumulated

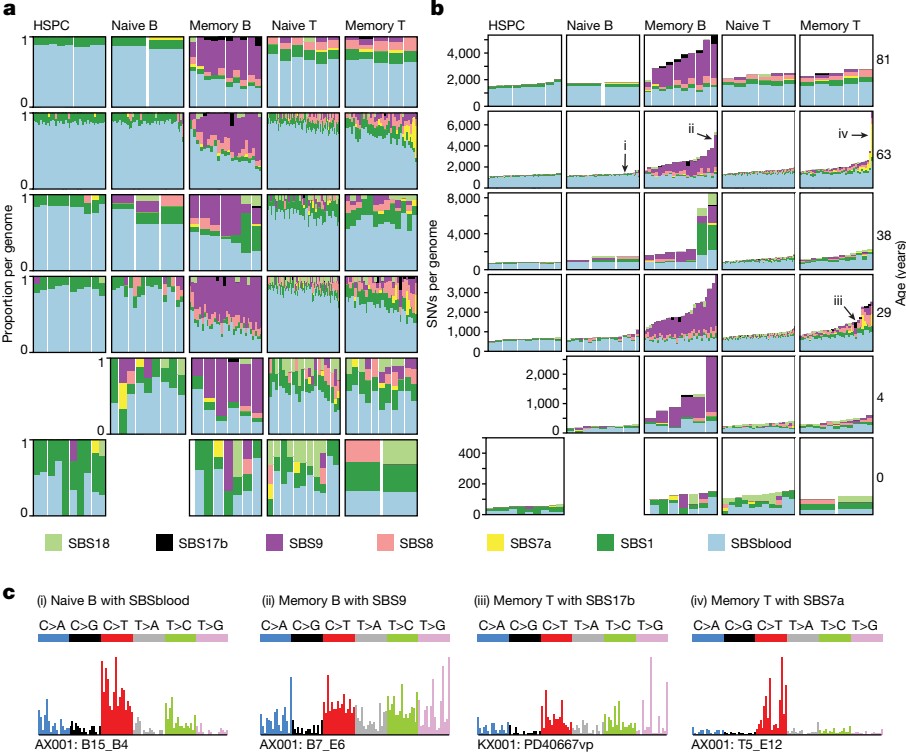

**Fig. 2 | Mutational processes in lymphocytes. a,b**, The proportion of SNVs (**a**) and SNV burden (**b**) per mutational signature in the different cell types. Each column represents one genome. For each genome, signatures with a 90% posterior interval lower bound of less than 1% are excluded. **c**, Mutational spectra of genomes of colonies derived from single cells enriched in the specified mutational signature. The specific genome plotted is numbered in **b**. Trinucleotide contexts on the *x*-axis represent 16 bars within each substitution class, divided into 4 sets of 4 bars, grouped by the nucleotide 5′ to the mutated base, and within each group by the 3′ nucleotide (in the order A, C, G, T).

base substitutions at approximately 16 SNVs per cell per year (95% confidence interval 13–19), similar to previous estimates[10,12]. Naive and memory B cells showed broadly similar rates of mutation accumulation (naive B cells: 15 SNVs per cell per year, 95% confidence interval 12–18; memory B cells: 17 SNVs per cell per year, 95% confidence interval 6–28). However, T cells had higher mutation rates (naive T cells: 22 SNVs per cell per year, 95% confidence interval 19–25; memory T cells: 25 SNVs per cell per year, 95% confidence interval 17–32). Overall, this suggests that there are clock-like mutational processes adding mutations at constant rates, with different rates in each lymphocyte subset.

Additionally, there was a significant increase in the burden of base substitutions in lymphocytes that could not be explained by age, especially for memory cells. Compared with HSPCs, naive B and T cells had an average of 110 (95% confidence interval 5–216) and 59 (95% confidence interval −35 to 153) extra SNVs per cell, respectively, beyond the effects of age. Memory B and T cells had an even more pronounced excess of mutations, carrying an average of 1,034 (95% confidence interval 604–1,465) and 277 (95% confidence interval 5–549) more SNVs per cell than HSPCs, respectively. This extra burden of base substitutions presumably represents variants acquired during differentiation: approximately 100 from HSPC to naive cell and hundreds to thousands from naive to memory cell.

We found that the variance in mutation burden across cells also showed a massive increase with differentiation. Thus, compared to a s.d. of 70 SNVs per cell for HSPCs within a given donor, the values estimated for memory B and T cells were 820 SNVs per cell and 592 SNVs per cell respectively ($P < 10^{-16}$ for heterogeneity of variance across cell types). This cell-to-cell variability within a donor considerably outweighed the between-person s.d., which we estimated at 60 SNVs per cell.

Indels accumulated at an average of 0.7 per cell per year in HSPCs (95% confidence interval 0.5–0.9), while lymphocytes had higher indel rates (naive B cells: 0.8 per cell per year, 95% confidence interval 0.6–1.0; naive T cells: 1.1, 95% confidence interval 0.9–1.2; memory B cells: 0.8, 95% confidence interval 0.4–1.3; memory T cells: 1.0, 95% confidence interval 0.7–1.2; Extended Data Fig. 3a).

Somatic mutations can confer a selective advantage on normal cells, driving clonal expansions. Global measures of the strength of positive selection can be obtained by estimating the excess of non-synonymous mutations (*N*) compared to selectively neutral synonymous (*S*) mutations[13] (d*N*/d*S* ratio, with d*N*/d*S* = 1 denoting neutrality). Exome-wide, excluding immunoglobulin regions, we estimated the dN/dS ratio in lymphocytes to be 1.12 (95% confidence interval 1.06–1.19). This implies that positive selection shapes clonal competition in lymphocytes, with approximately 11% (95% confidence interval 6–15%) of non-synonymous mutations conferring a selective advantage (Extended Data Fig. 3b). At a single-gene level, *ACTG1* was the only gene significant with a false-discovery rate of less than 1% ($q = 5 \times 10^{-3}$)—this gene is recurrently mutated in the plasma cell malignancy multiple myeloma[14,15].

## Mutational signatures

To determine whether the excess mutations observed in lymphocyte subsets were owing to a specific mutational process, we inferred mutational signatures across lymphocyte compartments (Fig. 2). Similar to HSPCs, the vast majority of mutations in naive B and T cells were derived from two mutational signatures. One of these—SBS1—is caused by spontaneous deamination of methylated cytosines, and accounted for 14% of mutations in HSPCs and naive B and T cells. Nearly all the remaining somatic mutations in these cellular compartments had the typical signature of endogenous mutations in HSPCs[10,11], which we term SBSblood (Extended Data Fig. 4a). The burden of both

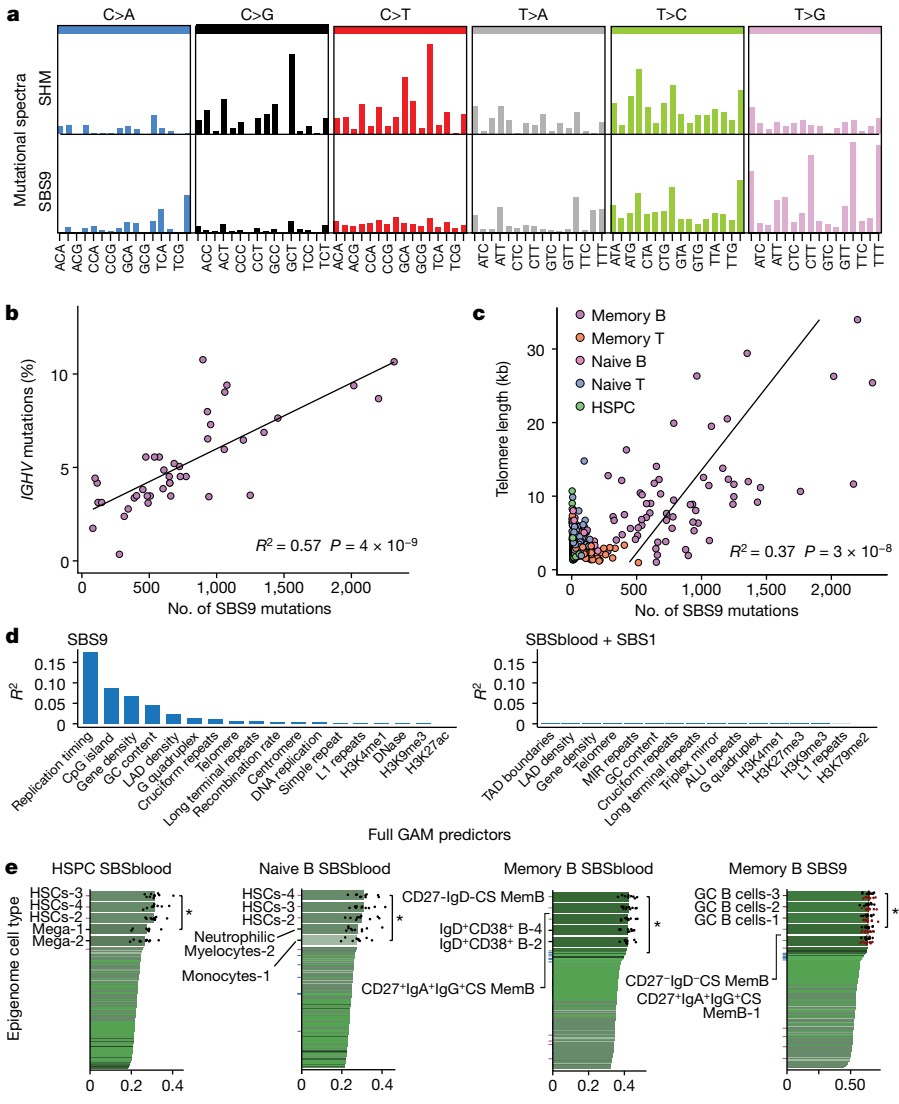

**Fig. 3 | Correlation of SBS9 with genomic attributes and timing of mutational processes. a**, Mutational spectra of the SBS9 and SHM signatures. Trinucleotide contexts on the *x*-axis represent 16 bars within each substitution class, divided into 4 sets of 4 bars, grouped by the nucleotide 5′ to the mutated base, and within each group by the 3′ nucleotide. The *y*-axis shows the number of mutations in each class. **b**, The number of SBS9 mutations genome-wide and the percentage of bases in *IGHV* that are mutated in the productive rearrangement of memory B cells. The line represents the linear regression estimate of the correlation. **c**, Number of SBS9 mutations versus telomere length per genome, coloured by cell type. The regression line is for memory B cell. **d**, Explanatory power of each significant genomic feature in the generalized additive model (GAM), expressed as the *R*² of the individual GAM

for predicting number of SBS9 mutations (left) or number of SBSblood or SBS1 mutations (right) per 10-kb window. LAD, lamina-associated domain. **e**, Performance of prediction of genome-wide mutational distribution attributable to particular mutational signatures from histone marks of 149 epigenomes representing distinct blood cell types and different phases of development (numbers after cell types on *y*-axis indicate replicates); ticks are coloured according to the epigenetic cell type (purple, HSC; blue, naive B cell; grey, memory B cell; maroon, GC B cell); black points depict values from tenfold cross-validation; *P*-values for comparison of the tenfold cross-validation values by two-sided Wilcoxon test. CS, class switched; GC, germinal centre; HSC, hematopoietic stem cell; Mem, memory; Mega, megakaryocyte.

signatures correlated linearly with age (Extended Data Fig. 4b,c), suggesting that they represent clock-like endogenous mutational processes.

For memory B and T cells, the absolute numbers of mutations attributed to these two endogenous signatures were broadly similar to those seen in naive B and T cells (Fig. 2b). The hundreds to thousands of extra mutations seen in memory B and T cells derived from additional mutational signatures: SBS7a, SBS8, SBS9 and SBS17b. Whereas signatures SBS8 and SBS9 show correlations with age, SBS7a and SBS17a do not, consistent with them being sporadic. SBS7a and SBS17b probably represent exogenous mutational processes, whereas SBS9 is associated with differentiation, as discussed below.

## Exogenous mutational signatures

SBS7a is the canonical signature of ultraviolet light damage, the predominant mutational process in melanoma[16] and normal skin[17]. The signature that we extracted from memory cells matches the features of SBS7a, with a predominance of C>T substitutions in a dipyrimidine context, transcriptional strand bias and a high rate of CC>TT dinucleotide substitutions (Fig. 2c and Extended Data Fig. 5). We found a substantial contribution of SBS7a (more than 10% of mutations; mean = 757 per cell, range 205–2,783) and CC>TT dinucleotide substitutions in 9 out of 100 memory T cells. Notably, memory cells with high levels of SBS7a mutations had significantly shorter telomeres than other memory

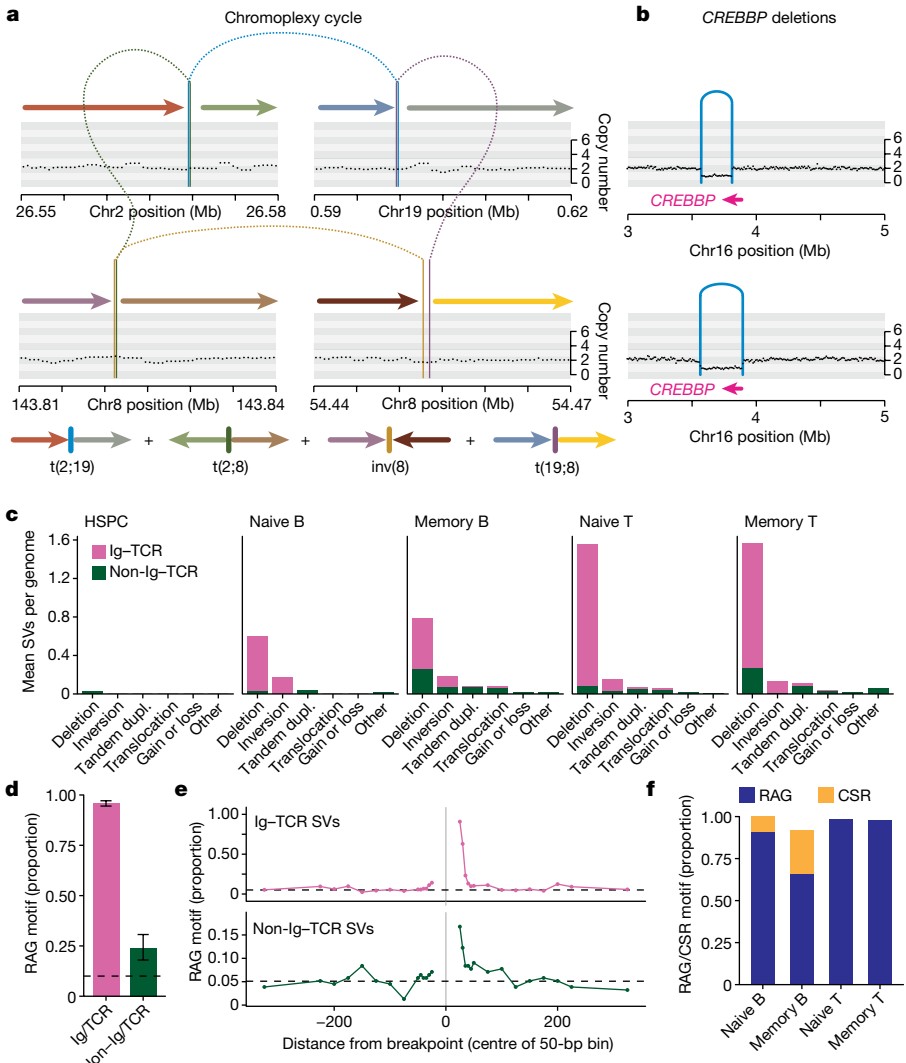

**Fig. 4 | Structural variation burden and off-target RAG-mediated deletion.** **a**, Top, chromoplexy cycle (sample PD40667sl, donor KX002). Black points represent the corrected read depth along the chromosome and arcs denote structural variants. Bottom, the final genomic configuration of the four derivative chromosomes is shown as coloured arrows. **b**, *CREBBP* deletions (samples PD40521po, donor KX001 and BMH1_PlateB1_E2, donor AX001). **c**, Burden of structural variants per cell type. Dupl., duplication. **d**, The proportion of deletions with an RSS (RAG) motif within 50 bp of the breakpoint for Ig–TCR (0.96) and non-Ig–TCR (0.24) regions. The black dashed line represents the genomic background rate of RAG motifs. Error bars represent 95% bootstrap confidence intervals. $n$ = 889 Ig–TCR structural variants and 253 non-Ig–TCR structural variants. **e**, Proportion of deletions with an RSS (RAG) motif as a function of distance from the breakpoint, with a positive distance representing bases interior to the deletion, and a negative value representing bases exterior to the breakpoint. The black dashed line represents the genomic background rate of RAG motifs. **f**, The proportion of deletions with an RSS (RAG) or switch (CSR) motif.

T cells ($P$ = 0.01, Fisher's method; Extended Data Fig. 5b), indicative of increased proliferation. As UVB radiation only penetrates human skin[18] to a depth of 10–50 μm, the most plausible source of these SBS7a mutations is UV exposure during skin residency.

A second unexpected signature in memory cells was SBS17. This signature has been observed in cancers of the stomach and oesophagus and occasionally in B and T cell lymphomas[16]. This signature, characterized by T>G mutations in a TpT context (the underline indicates the mutated base), accounted for more than 10% of mutations (4× s.d. above the mean) in 3 out of 74 memory B and 1 out of 100 memory T cells. SBS17 has been linked to 5-fluorouracil chemotherapy in metastatic cancers[19,20], but its occurrence in primary oesophageal and gastric cancers (as well as our samples here) is independent of treatment. If its incidence in upper gastrointestinal tract cancers is caused by some unknown local mutagen, then the presence of SBS17 in memory cells may be evidence of a specific microenvironmental exposure associated with tissue residency in gastrointestinal mucosa.

## Signatures of the germinal centre

SHM at heavy and light chain immunoglobulin regions followed the expected mutational signature (Fig. 3a), with the productive rearrangement showing more mutations than non-recombined alleles (Extended Data Fig. 6a–c). However, as reported for lymphoid malignancies[5], off-target mutations with the SBS9 signature in memory B cells had a different spectrum to SHM mutations, characterized by mutations at A:T base pairs in a TpW context (Fig. 3a), and different distribution across the genome (Extended Data Fig. 6d). SBS9 accounted for 42% of mutations (mean = 780 mutations per cell) in memory B cells, sometimes tripling the baseline mutation burden.

The number of SBS9 mutations genome-wide showed a strong linear correlation with the SHM rate (the percentage of the productive *IGHV* gene that was mutated), despite their different spectra ($R^2$ = 0.57, $P$ = 4 × 10$^{-9}$, linear regression; Fig. 3b). The density of mutations was

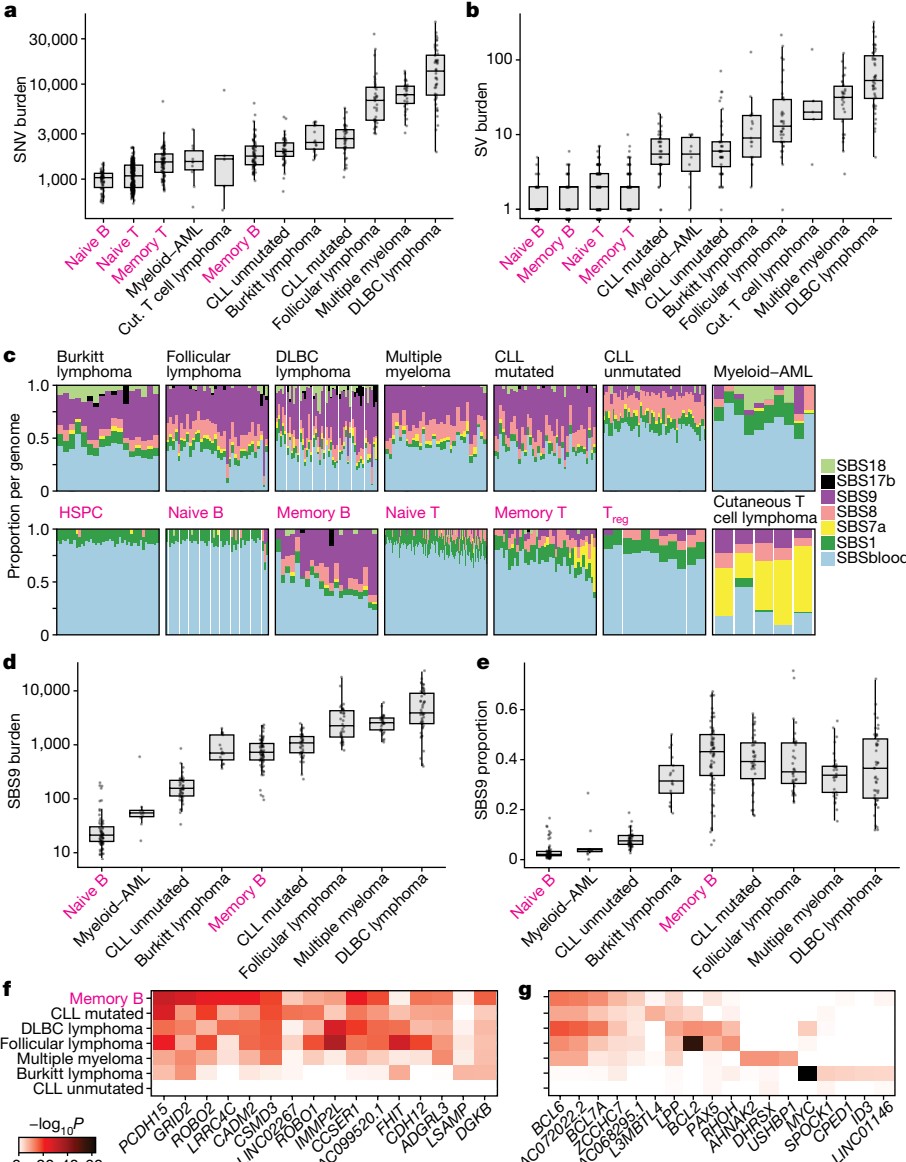

**Fig. 5 | Comparison of mutational patterns with malignancy. a,b,** SNV (**a**) and structural variation (SV) burden (**b**) by normal cell type or malignancy. The box shows the interquartile range and the centre line shows the median. Whiskers extend to the minimum of either the range or 1.5× the interquartile range. Normal lymphocytes (magenta) exclude paediatric samples. AML, acute myeloid leukaemia; CLL, chronic lymphocytic leukaemia; Cut, cutaneous; DLBC, diffuse large B cell. **c,** The proportion of mutational signatures per genome. For each genome, signatures with a 90% confidence interval lower bound of less than 1% are excluded. Normal lymphocytes (labelled in magenta) are from donor AX001. T_reg, T regulatory cells. **d,e,** SBS9 burden (**d**) and proportion (**e**) by cell type or malignancy. The box shows the interquartile

range and the centre line shows the median. Whiskers extend to the minimum of either the range or 1.5× the interquartile range. **f,g,** Heat map showing the level of enrichment of SBS9 (**f**) and SHM (**g**) signatures near frequently mutated genes for that signature compared with the whole genome. Number of structural variants per group: B cell: 145, T cell: 841, ALL: 523, Burkitt lymphoma: 305, CLL mutated: 252, CLL unmutated: 440, cutaneous T cell lymphoma: 204, DLBC lymphoma: 3,754, follicular lymphoma: 1,095. **a,b,d,e,** Number of genomes per group: naive B: 68, memory B: 68, naive T: 332, memory T: 87, Burkitt lymphoma: 17, CLL mutated: 38, CLL unmutated: 45, cutaneous T cell lymphoma: 5, DLBC lymphoma: 47, follicular lymphoma: 36, multiple myeloma: 30, myeloid–AML: 10.

270,000-fold greater at the *IGHV* locus than for SBS9 mutations genome-wide, confirming the precise targeting of SHM to antibody regions. Nonetheless, the genome is large, and even this high degree of mutational targeting means that every on-target *IGHV* mutation is accompanied by an average of 18 SBS9 mutations elsewhere in the genome.

Another feature of the germinal centre reaction is increased telomerase activity in B cells[21,22]. We estimated telomere lengths from the genome sequencing data for our dataset. Telomere lengths in HSPCs, T cells and naive B cells decreased by 30–50 bp per year over a lifetime[23–25], consistent with cell divisions occurring every 6–24 months (Extended Data Fig. 7a). By contrast, telomere lengths in memory B cells

were longer, more variable and actually increased with age (excluding tonsil samples; $R^2 = 0.13$, $P = 3 \times 10^{-3}$, linear regression). Telomere lengths also correlated linearly with the number of SBS9 mutations genome-wide ($R^2 = 0.37$, $P = 3 \times 10^{-8}$; Fig. 3c). This correlation supports a hypothesis of lengthening telomeres and occurrence of off-target SBS9 mutations during the germinal centre reaction.

## A replicative-stress model of SBS9

The cytosine deaminase AID initiates on-target SHM at immunoglobulin loci, which generates damage (and consequent mutation) at C:G base

pairs. On-target mutations at A:T base pairs during SHM arise through errors introduced during translesion bypass of AID-deaminated cytosines by polymerase η[26], which has an error spectrum weighted towards a TpW context[27]. As has been noted in lymphoid malignancies[5,16], SBS9 has a different spectrum from on-target, AID-mediated SHM, something we also observe in normal lymphocytes. In particular, SBS9 has a paucity of mutations at C:G base pairs and an enrichment of T mutations in TpW context (Fig. 3a), which makes the role of AID unclear because it specifically targets cytosines. The genome-wide distribution of off-target AID-induced deamination has been measured directly[28], and shows a predilection for highly transcribed regions with active chromatin marks, which tend to be early replicating.

To explore whether genomic regions with high SBS9 burden show the same distribution, we used general additive models to predict SBS9 burden from 36 genomic features, including gene density, chromatin marks and replication timing across 10-kb genome bins. After model selection, 18 features were included in the regression ($R^2 = 0.20$; Fig. 3d and Supplementary Table 4). Replication timing is by far the strongest predictor, with increased mutation density in late-replicating regions, individually accounting for 17% of the variation in the genomic distribution of SBS9 (Extended Data Fig. 7b). By contrast, replication timing accounted for only 0.6% of variation in density of SBSblood or SBS1 mutations in memory B cells and 0.1% in HSPCs. The next 4 strongest predictors of SBS9 distribution were all broadly related to inactive versus active regions of the genome (distance from CpG islands, gene density, GC content and LAD density: individual $R^2$ values of 0.09, 0.07, 0.05, and 0.02, respectively). For each variable, mutation density increased in the direction of less active genomic regions—this is in contradistinction to AID-induced deamination, which occurs in actively transcribed regions[28].

Together, our data demonstrate that SBS9 accumulates during the germinal centre reaction, evidenced by its tight correlation with both on-target SHM and telomere lengthening. However, the relative sparsity of mutations at C:G base pairs and the distribution of SBS9 to late-replicating, repressed regions of the genome make it difficult to argue that AID is involved. Instead, we hypothesize that SBS9 arises from polymerase η bypass of other background DNA lesions induced by the high levels of replicative and oxidative stress experienced by germinal centre B cells. Normally, mismatch repair and other pathways would accurately correct such lesions, but the high expression of polymerase η in germinal centre cells[29] provides the opportunity for error-prone translesion bypass to compete. The enrichment of SBS9 in late-replicating, gene-poor, repressed regions of the genome—regions where mismatch repair is typically less active[30,31]—would be consistent with this as a model of SBS9 mutation.

## Epigenetic marks reveal mutation timing

Among human cell types, lymphocytes are unusual for passing through functionally distinct, long-lived differentiation stages with ongoing proliferative potential. Since variation in mutation density across the genome is shaped by chromatin state, a cell's specific distribution of somatic mutations provides a record of the past epigenetic landscape of its ancestors back to the fertilized egg[32,33]. We thus hypothesized that the distribution of clock-like signatures would inform on the cell types present in a given cell's ancestral line of descent. By contrast, the distribution of sporadic or episodic signatures can inform on the differentiation stage exposed to that particular mutational process.

We compared the distribution of somatic mutations across the genome with 149 epigenomes representing 48 distinct blood cell types and differentiation stages. Mutations resulting from the clock-like signature SBSblood in HSPCs correlated best with histone marks from haematopoietic stem cells ($P = 0.002$, Wilcoxon test; Fig. 3e), consistent with mutation accumulation in undifferentiated cells. Notably, SBSblood mutational profiles in naive B cells also correlated better

with the epigenomes of haematopoietic stem cells than naive B cells ($P = 0.004$; Fig. 3e). This implies that the majority of SBSblood mutations in naive B cells were acquired pre-differentiation, consistent with ongoing production of these cells from the HSPC compartment throughout life and a relatively short-lived naive B differentiation state. By contrast, SBSblood mutations in naive T cells mapped best to the epigenomes of CCR7[+]CD45RO[−]CD25[−]CD235[−] naive T cells ($P = 0.049$; Extended Data Fig. 8), consistent with a large, long-lived pool of naive T cells generated in the thymus during early life. For memory B cells, SBSblood most closely correlated with histone marks from that cell type and not earlier differentiation stages ($P = 0.02$; Fig. 3e), suggesting that the majority of their lineage has been spent as a memory B cell.

For the sporadic mutational processes, SBS9 mutations most closely correlated with germinal centre B cell epigenomes ($P = 0.049$; Fig. 3e). This is consistent with our finding of a correlation between SBS9 and other processes associated with germinal centres (SHM and telomere lengthening), providing further evidence that SBS9 arises as a by-product of the germinal centre reaction. For SBS7a, the signature of ultraviolet light exposure seen in memory T cells, the genomic distribution is more tightly correlated with epigenomes of differentiated T cells than naive T cells (Extended Data Fig. 8), supporting the hypothesis that SBS7a mutations accumulate in differentiated T cells.

## Structural variants

Both V(D)J recombination and CSR are associated with off-target structural variants in human lymphoid malignancies[2,3,7], but rates and patterns of structural variants have not been studied in normal human lymphocytes. We found 1,037 structural variants across 635 lymphocytes, 85% of which occurred in immunoglobulin or TCR (Ig–TCR) regions (Extended Data Fig. 9). We identified fewer than the 2 expected on-target V(D)J recombination events per lymphocyte, suggesting that the sensitivity for structural variants in these regions in our experiments is approximately 62%.

Excluding Ig–TCR gene regions, B and T cells carried more structural variants than HSPCs, with 103 out of 609 (17%) of lymphocytes having at least one off-target structural variant (compared with a single structural variant in 82 HSPCs; $P = 9 × 10^{-5}$, Fisher's exact test). Memory B and T cells had higher non-Ig–TCR structural variant burdens than their respective naive subsets (27% in memory B cells versus 5% in naive B cells; 25% in memory T cells versus 15% in naive T cells; $P = 1 × 10^{-5}$). Although there were occasional instances of more complex abnormalities, including chromoplexy (Fig. 4a) and cycles of templated insertions[34], most non-Ig–TCR structural variants were deletions (49%), several of which affected genes mutated in lymphoid malignancies (Fig. 4b and Supplementary Table 5).

V(D)J recombination is mediated by RAG1 and RAG2 cutting at a recombination signal sequence (RSS) DNA motif comprising a heptamer and nonamer with an intervening spacer. Twenty-four per cent of non-Ig–TCR and 96% of Ig–TCR structural variants had a full RSS motif or the heptamer within 50 bp of a breakpoint (Fig. 4c,d). Accounting for the baseline occurrence of these motifs using genomic controls, we estimate that 12% of non-Ig–TCR and 84% of Ig–TCR structural variants were RAG-mediated, especially deletions (around 15% of non-Ig–TCR deletions). As expected, the RSS motif was typically internal to the breakpoint (62% and 91% for non-Ig–TCR and Ig–TCR structural variants). We observed a rapid decay in the enrichment of RAG motifs with distance from breakpoints, reaching background levels within about 100 bp (Fig. 4e). During V(D)J recombination, the TdT protein adds random nucleotides at the dsDNA breaks—this also occurs in off-target structural variants, with RAG-mediated events enriched for insertions of non-templated sequence at the breakpoint (44% and 88% for non-Ig–TCR and Ig–TCR structural variants, respectively, versus 21% of off-target structural variants without an RSS motif; $P = 9 × 10^{-3}$, Fisher's exact test).

CSR is achieved through AID cytosine deamination at WGCW clusters, deleting IgH constant region genes and changing the antibody isotype. As expected, on-target CSR was enriched in memory (76%) compared with naive B cells (12%; Fig. 4f and Supplementary Table 6). By contrast, none of the non-Ig–TCR structural variants had CSR AID motif clusters, suggesting that CSR is exquisitely targeted.

## Comparison with malignancy

A long-standing controversy in cancer modelling is whether tumours require additional mutational processes to acquire sufficient driver mutations for oncogenic transformation[35]. In many solid tissues, cancers have higher mutation burdens than normal cells from the same organ[36,37], but myeloid leukaemias do not[9]. To address this question in lymphoid malignancies, we compared genomes from normal B and T cells to eight blood cancers[15,38,39], which had similar distributions of effective sequencing coverage (Extended Data Fig. 9c). SNV burdens of follicular lymphoma, diffuse large B cell lymphoma and multiple myeloma were considerably higher than those of normal lymphocytes (Fig. 5a,b). By contrast, point mutation burdens observed in Burkitt lymphoma, mutated or unmutated chronic lymphocytic leukaemia and acute myeloid leukaemia were well within the range of those in normal lymphocytes. All lymphoid malignancies showed higher rates of structural variation than normal cells.

The increased point mutation burden could arise from increased activity of mutational processes already present in normal cells, or the emergence of distinct, cancer-specific mutational processes. The vast majority of mutations present across all B cell malignancies could be attributed to the same mutational processes active in normal memory B cells, and in broadly similar proportions (Fig. 5c–e). Cutaneous T cell lymphomas carried similar numbers of mutations attributable to ultraviolet light as the high-SBS7a memory cells (Extended Data Fig. 5c). These data emphasize that the processes generating point mutations in normal lymphocytes can generate sufficient somatic variants for progression towards many types of lymphoid malignancy.

A feature of somatic mutations in B cell lymphomas is clustering of off-target SHMs in highly expressed genes. For both SBS9 (Fig. 5f) and off-target SHMs (Fig. 5g), we found considerable overlap in genes with elevated mutation rates. For example, *BCL6*, *BCL7A* and *PAX5* exhibited enrichment of mutations with the SHM signature in both normal and post-germinal malignant lymphocytes. Similarly, out of the 100 genes most enriched for SBS9 in normal memory B cells, 64% were also SBS9-enriched (top 1%) in at least 3 of the 5 post-germinal malignancies.

About 10% of normal lymphocytes have a non-Ig–TCR RAG-mediated structural variant, accounting for 24% of off-target rearrangements. Across lymphoid malignancies, acute lymphoblastic leukaemia had similarly high proportions of RAG-mediated events, but in much higher numbers, as reported previously[2,3] (Extended Data Fig. 10a). For other lymphoid malignancies, although the proportions were low, the absolute numbers of RAG-mediated structural variants (≥0.5 per lymphoma) were broadly comparable to those seen in normal lymphocytes (Extended Data Fig. 10b). This suggests that malignant transformation of lymphocytes is associated with the emergence of cancer-specific genomic instability, generating a genome with considerably more large-scale rearrangement.

## Discussion

Positive selection acting on somatic mutations in lymphocytes is more pervasive than negative selection, suggesting that clonal expansions of individual lymphocytes are the evolutionary trade-off for physiological genome editing. Lymphoid cancers are clearly one consequence—that mutation burdens and signatures of normal lymphocytes match those seen in lymphoid malignancies argues that off-target mutagenesis is sufficient to transform occasional lymphocytes. For more than 50 years, there has been speculation that driver mutations could underpin autoimmune diseases[40–42], with recent data showing driver mutations in lymphocytes responsible for vasculitis associated with Sjögren's disease[43]. Our data show, first, that mutation rates are high enough to generate considerable genetic diversity among normal lymphocytes, and second, that selective pressures favour clonal expansion of individual lymphocytes.

Unique among human cell types, a lymphocyte experiences long periods of its life in diverse microenvironments such as marrow, thymus, lymph node, skin or mucosa. Given that lymphocytes divide[44] every 3–24 months, data supported by our estimates of telomere attrition, mutation rates during these maintenance phases would presumably be 5–50 per cell division. These stages are interspersed with short-lived bursts of differentiation, each of which is associated with proliferation and/or programmed genome engineering to improve antigen recognition, contributing additional mutations. The considerably greater cell-to-cell variation than person-to-person variation suggests that lifelong environmental forces (such as infections, inflammation and skin residency) are stronger influences on lymphocyte genomes than the inherited variation in mutation rates. The signatures of these mutations reflect both the unintended by-products of immunological diversification and exposure to exogenous mutagens; their genomic distribution reflects the chromatin landscape of the cell at the time the mutational process was active.

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

## Methods

### Samples

Human blood mononuclear cells (MNCs) were obtained from four sources: (1) bone marrow, spleen and peripheral blood taken with written informed consent (provided by next-of-kin) from three deceased transplant organ donors (KX001, KX002, KX003) recruited from Cambridge University Hospitals NHS Trust, Addenbrooke's Hospital (by Cambridge Biorepository for Translational Medicine, Research Ethics Committee approval 15/EE/0152), (2) peripheral blood taken with written informed consent from one patient (AX001) recruited from Addenbrooke's Hospital (approval 07-MRE05-44), (3) tonsil taken with written informed consent from guardians of two patients (TX001, TX002) recruited from Addenbrooke's Hospital (approval 07-MRE05-44), and (4) one cord blood (CB001) collected with written informed consent from guardian by StemCell Technologies (catalogue (cat.) no. 70007) (Supplementary Table 2). All donors were haematopoietically normal and healthy. Donor KX002 had a history of Crohn's disease and treatment with Azathioprine. Patients TX001 and TX002 had a history of tonsillitis. MNCs from (1), (2) and (3) were extracted using Lymphoprep (Axis-Shield), depleted of red blood cells using RBC lysis buffer (BioLegend) and frozen viable in 10% DMSO. Cord blood MNCs (4) were received frozen and then selected on the basis of CD34 expression using the EasySep human whole-blood CD34 positive-selection kit (Stemcell Technologies) as per the manufacturer's instructions, with the CD34$^+$ fraction used for HSPC cultures and the CD34$^-$ fraction used for lymphocyte cultures. Additional peripheral blood MNCs from (1) also underwent CD34 positive selection and was used for HSPC cultures.

### Flow cytometry

MNC samples were sorted by flow cytometry at the NIHR Cambridge BRC Cell Phenotyping Hub on AriaIII or Aria-Fusion cell sorters into naive B cells (CD3$^-$CD19$^+$CD20$^+$CD27$^-$CD38$^-$IgD$^+$), memory B cells (CD3$^-$CD19$^+$CD20$^+$CD27$^+$CD38$^-$IgD$^-$), naive T cells (CD3$^+$CD4/CD8$^+$ CCR7$^+$CD45RA$^{high}$), memory T cells (CD3$^+$CD4/CD8$^+$CD45RA$^-$), regulatory T cells (Tregs: CD3$^+$CD4$^+$CD25$^{high}$CD127$^-$) and HSPCs (CD3$^-$CD19$^-$ CD34$^+$CD38$^-$CD45RA$^-$) (Supplementary Fig. 1). HSPCs from AX001 included HSCs (CD34$^+$CD38$^-$) and progenitors (CD34$^+$CD38$^+$CD10$^{-/dim}$). The antibody panels used are as follows: lymphocytes (excluding Tregs): CD3-APC, CD4-BV785, CD8-BV650, CD14-BV605, CD19-AF700, CD20-PEDazzle, CD27-BV421, CD34-APC-Cy7, CD38-FITC, CD45RA-PerCP-Cy5.5, CD56-PE, CCR7-BV711, IgD-PECy7, Zombie-Aqua; T$_{reg}$ cells: CD3-APC, CD4-BV785, CD8-BV650, CD19-APC-Cy7, CD45RA-PerCP-Cy5.5, CD56-PE, CCR7-FITC, CD25-PECy5, CD127-PECy7, CD69-AF700, CD103-BV421, CCR9-PE, Zombie-Aqua; HSPCs (excluding AX001): CD3-FITC, CD90-PE, CD49f-PECy5, CD38-PECy7, CD33-APC, CD19-A700, CD34-APC-Cy7, CD45RA-BV421, Zombie-Aqua; HSPCs (AX001): CD38-FITC, CD135-PE, CD34-PE-Cy7, CD90-APC, CD10-APC-Cy7, CD45RA-V450, Zombie-Aqua. Details of the antibody panels used are in Supplementary Table 11. Cells were either single-cell sorted for liquid culture into 96-well plates containing 50 µl cell-type-specific expansion medium, or (for AX001 HSPCs) bulk-sorted for MethoCult plate-base expansion. Plotting of the fluorescence-activated cell sorting data was performed with FlowJo and FCS Express.

### In vitro liquid culture expansion

We designed novel protocols to expand B and T cells from single cells into colonies of at least 30 cells. Detailed step-by-step descriptions of the protocols are provided in Supplementary Information. The B cell expansion medium was composed of 5 µg ml$^{-1}$ Anti-IgM (Stratech Scientific), 100 ng ml$^{-1}$ IL-2, 20ng ml$^{-1}$ IL-4, and 50 ng ml$^{-1}$ IL-21 (PeproTech EC), 2.5 ng ml$^{-1}$ CD40L-HA (Bio-Techne) and 1.25 µg ml$^{-1}$ HA Tag (Bio-Techne), in Advanced RPMI 1640 Medium (ThermoFisher Scientific) with 10% fetal bovine serum (ThermoFisher Scientific),

1% penicillin/streptomycin (Sigma-Aldrich), and 1% L-glutamine (Sigma-Aldrich). The T cell expansion medium was composed of 12.5 µl ml$^{-1}$ ImmunoCult CD3/CD28 (STEMCELL Technologies) and 100 ng ml$^{-1}$ IL-2 and 5 ng ml$^{-1}$ IL-15 (PeproTech EC), in ImmunoCult-XF T Cell Expansion Medium (STEMCELL Technologies) with 5% fetal bovine serum (ThermoFisher Scientific) and 0.5% penicillin/streptomycin (Sigma-Aldrich). Twenty-five microlitres of fresh expansion medium was added to each culture every 3–4 days. Colonies (30–2,000 cells per colony) were collected either manually or robotically using a CellCelector (Automated Lab Solutions) approximately 12 days after sorting (depending on growth).

Sorted HSPCs from donors KX001, KX002, KX003 and CB001 were expanded from single cells into colonies of 200–100,000 cells in Nunc 96-well flat-bottomed TC plates (ThermoFisher Scientific) containing 100 µl supplemented StemPro medium (Stem Cell Technologies) (MEM medium). MEM medium contained StemPro Nutrients (0.035%) (Stem Cell Technologies), L-Glutamine (1%) (ThermoFisher Scientific), Penicillin-Streptomycin (1%) (ThermoFisher Scientific) and cytokines (SCF: 100 ng ml$^{-1}$; FLT3: 20 ng ml$^{-1}$; TPO: 100 ng ml$^{-1}$; EPO: 3 ng ml$^{-1}$; IL-6: 50 ng ml$^{-1}$; IL-3: 10 ng ml$^{-1}$; IL-11: 50 ng ml$^{-1}$; GM-CSF: 20 ng ml$^{-1}$; IL-2: 10 ng ml$^{-1}$; IL-7: 20 ng ml$^{-1}$; lipids: 50 ng ml$^{-1}$) to promote differentiation towards myeloid–erythroid–megakaryocyte (MEM) and natural killer cell lineages. Manual assessment of colony growth was made at 14 days. Colonies were topped up with an additional 50 µl MEM medium on day 15 if the colony was ≥1/4 the size of the well. Following 21 ± 2 days in culture, colonies were selected by size criteria. Colonies ≥3,000 cells in size were collected into a U-bottomed 96-well plate (ThermoFisher Scientific). Plates were then centrifuged (500$g$ for 5 min), medium was discarded, and the cells were resuspended in 50 µl PBS prior to freezing at −80 °C. Colonies less than 3,000 cells but greater than 200 cells in size were collected into 96-well skirted Lo Bind plates (Eppendorf) and centrifuged (800$g$ for 5 min). Supernatant was removed to 5–10 µl using an aspirator prior to DNA extraction on the fresh cell pellet. Sorted HSPCs from donor AX001 were plated onto CFC medium MethoCult H4435 (STEMCELL Technologies) and colonies were picked following 24 days in culture.

### Whole-genome sequencing of colonies

DNA was extracted from 717 colonies with Arcturus PicoPure DNA Extraction Kit (ThermoFisher Scientific), with the exception of larger HSPC colonies which were extracted using the DNeasy 96 blood and tissue plate kit (Qiagen) and then diluted to 1–5 ng. DNA was used to make Illumina sequencing libraries using a custom low-input protocol[45]. We performed whole-genome sequencing using 150 bp paired-end sequencing reads on an Illumina XTen platform, to an average depth of 20× per colony. Sequence data were mapped to the human genome reference GRCh37d5 using the BWA-MEM algorithm.

### Variant calling

We called all classes of variants using validated pipelines at the Wellcome Sanger Institute. SNVs were called using the program CaVEMan[46], insertion/deletions (indels) using Pindel[47], structural variants using BRASS[48] and copy number variants (CNVs) using ASCAT[49]. In order to recover all mutations, including high frequency ones, we used an in silico sample produced from the reference genome rather than use a matched normal for the CaVEMan, Pindel, and BRASS analyses. Germline mutations were removed after variant calling (see below). For the ASCAT analysis we elected one colony (arbitrarily chosen) to serve as the matched normal.

Variants were filtered to remove false positives and germline variants. First, variants with a mean VAF greater than 40% across colonies of an individual were probably germline variants and were removed. To remove remaining germline variants and false positives, we exploited the fact that we have several, highly clonal samples per individual. We performed a beta-binomial test per variant per individual, retaining

only SNVs and indels that were highly over-dispersed within an individual. For SNVs we also required that the variants be identified as significantly subclonal within an individual using the program Shearwater, and applied filters to remove artefacts resulting from the low-input library preparation. Detailed descriptions of the artefact filters were provided previously[45] and the complete filtering pipeline is made available on GitHub (https://github.com/MathijsSanders/SangerLCMFiltering). For both the beta-binomial filter and the Shearwater filter we observed bimodal distributions separating the data into low and high confidence variants. We made use of this feature, using a valley-finding algorithm (R package quantmod) to determine the p-value cut-offs, per individual. We genotyped each colony for the set of filtered somatic SNVs and indels (per respective individual), calling a variant present if it had a minimum VAF of 20% and a minimum of two alternate reads in that colony.

We estimated our sensitivity to detect SNVs using germline mutations as a truth set of heterozygous mutations. We called germline mutations by performing a one-sided exact binomial test of the sum of the alternate and sum of the total reads across colonies of an individual for each CaVEMan unfiltered variant (alternate hypothesis of proportion of successes less than 0.5 for autosomes and female X chromosomes, 0.95 for male sex chromosomes). A variant was called as germline on failure to reject the null at a false-discovery rate $q$-value of $10^{-6}$. We calculated sensitivity as the proportion of germline variants detected per colony.

We removed artefacts from the structural variant calls using AnnotateBRASS with default settings. The full list of statistics calculated and post-hoc filtering strategy was described in detail previously[36]. Somatic structural variants were identified as those shared by less than 25% of the colonies within an individual. Structural variants and CNVs were both subsequently manually curated by visual inspection.

## Mutation burden analysis

We found that sequencing depth was a strong predictor of mutation burden in our samples. Therefore, in order to more accurately estimate the mutation burden for each colony, we corrected the number of SNVs or indels (corrected separately) by fitting an asymptotic regression (function NLSstAsymptotic, R package stats) to mutation burden as a function of sequencing depth per colony. For this correction we used HSPC genomes (excepting the tonsil samples, for which naive B and T cells were used), as lymphocyte genomes are more variable in mutation burden, and included additional unpublished HSPC genomes to increase the reliability of the model[12]. Genomes with a mean sequencing depth of greater than 50× were omitted. The model parameters $b_0$, $b_1$ and lrc for each dataset for the model $y = b_0 + b_1 \times (1 - \exp(-\exp(\text{lrc}) \times x))$ are in Supplementary Table 7. Mutation burden per colony was adjusted to a sequencing depth of 30.

We used a linear mixed-effects model (function lme, R package nlme) to test for a significant linear relationship between mutation burden and age, and for an effect of cell subset on this relationship (separately for SNVs and indels). Number of mutations per colony was regressed on age of donor and cell type as fixed effects, with interaction between age and cell type, donor by cell type as a random effect, weighted by cell type, and with maximum likelihood estimation.

## Detecting positive selection

In order to estimate an exome-wide rate of selection and to detect selection acting on specific genes we used the dndscv function of the dNdScv R package[13]. This program leverages mutation rate information across genes. As the elevated mutation rate seen with SHM may break the assumptions of the test, we excluded the immunoglobulin loci from these analyses (excluded GRCh37 regions: chr14:106304735–107283226, chr2:89160078–90274237, chr22:22385390–23263607). We performed the test for the following subsets of the data: all lymphocytes, naive B, memory B, naive T, memory T, all lymphocytes testing only cancer genes and all lymphocytes excluding cancer genes. Cancer genes were defined as the 566 tier 1 genes from the COSMIC Cancer Gene Census (https://cancer.sanger.ac.uk, downloaded 6 June 2018).

## Mutational signature analysis

We characterized per-colony mutational profiles by estimating the proportion of known and novel mutational signatures present in each colony. For comparison, we included in the analysis 223 genomes from 7 blood cancer types: Burkitt lymphoma, follicular lymphoma, diffuse large B cell lymphoma, chronic lymphocytic leukaemia (mutated), chronic lymphocytic leukaemia (unmutated), and acute myeloid leukaemia[38] and multiple myeloma[15]. We identified mutational signatures present in the data by performing signature extraction with two programs, SigProfiler[50] and hdp (https://github.com/nicolaroberts/hdp). We used the SigProfiler de novo results for the suggested number of extracted signatures. hdp was run without any signatures as prior, with no specified grouping of the data. These programs identified the presence of 9 mutational signatures with strong similarity (cosine similarity ≥ 0.85) to Cosmic signatures[16] SBS1, SBS5, SBS7a, SBS8, SBS9, SBS13, SBS17b, SBS18 and SBS19 (version 3).

Both SigProfiler and hdp also identified the same novel signature (cosine similarity = 0.93), which we term the blood signature or SBSblood. This signature is very similar to the mutational profile seen previously in HSPCs[10,11]. As the signature SBSblood co-occurs with SBS1 in HSPCs, leading to the potential for these signatures being merged into one signature, we further purified SBSblood by using the program sigfit[51] to call two signatures across our HSPC genomes, SBS1 and a novel signature, with the novel signature being the final SBSblood (Extended Data Fig. 4a and Supplementary Table 8). SBSblood was highly similar to both the hdp and SigProfiler de novo extracted signatures (cosine similarity of 0.95 and 0.94, respectively) and had similarity to the Cosmic v3 SBS5 signature (cosine similarity = 0.87). One hypothesis is that SBSblood is the manifestation of SBS5 mutational processes in the blood cell environment.

We estimated the proportion of each of the 10 identified mutational signatures using the program sigfit. From these results we identified three signatures (SBS5, SBS13 and SBS19) that were at nominal frequencies in the HSPC and lymphocyte genomes (less than 10% in each genome)- these were excluded from the analysis and the signature proportions were re-estimated in sigfit using the remaining 7 signatures: SBSblood, SBS1, SBS7a, SBS8, SBS9, SBS17b, SBS18 (Supplementary Table 8).

## Immunoglobulin receptor sequence analysis

In order to identify the immunoglobulin rearrangements, productive CDR3 sequences and per cent SHM for each memory B cell, we ran IgCaller[52], using a genome from the same donor (HSPC or T cell) as a matched normal for germline variant removal. We considered the SHM rate to be the number of variants identified by IgCaller in the productive *IGHV* gene divided by the gene length. For CSR calling, see Supplementary Information.

We estimated the number of mutations resulting from on-target (*IGHV* gene) SHM compared with those associated with SBS9. We first counted all *IGHV* variants identified by Caveman pre-filtering, as we found that standard filtering removes many SHM variants. We then estimated SBS9 burden as the proportion of SBS9 mutations per genome multiplied by the SNV burden. The SBS9 mutation rate per genome was the SBS9 burden divided by the 'callable genome' (genome size of 3.1 Gb minus an average of 383 kb excluded from variant calling).

## Distribution of germinal centre-associated mutations in B cells

We assessed the genomic distribution of the germinal centre-associated mutational signatures, SBS9 and the SHM signature, in memory B cells. We performed per-Mb de novo signature analyses with hdp (no a priori signatures), treating mutations across all normal memory B cells within a given Mb window as a sample. The extracted SHM signature

(Supplementary Table 8) had a cosine similarity of 0.96 to the spectrum of memory B cell mutations in the immunoglobulin gene regions, supporting the assumption that it is indeed the signature of SHM. In this analysis, SBSblood and SBS1 resolved as a single combined signature that we refer to in the genomic feature regression (below) as SBSblood/SBS1.

We estimated the per-gene enrichment of SBS9 and SHM signatures across normal memory B and malignant B cell genomes (Burkitt lymphoma, follicular lymphoma, diffuse large B cell lymphoma, chronic lymphocytic leukaemia, and multiple myeloma). We first used sigfit to perform signature attribution of the signatures found in memory B cells (from the main signature analysis; SBSblood, SBS1, SBS8, SBS9, SBS17b or SBS18) and the extracted SHM signature from the above 1-Mb hdp analysis, considering each 1-Mb bin a sample. We subsequently calculated a signature attribution per variant. Gene coordinates were downloaded from UCSC (gencode.v30lift37.basic.annotation.gene-only.genename.bed). We calculated the mean attribution of variants in a given gene, representing the proportion of variants attributable to a given signature. We estimated the enrichment of SBS9 or SHM over genomic background per gene per cell type as the $P$-value of individual $t$-tests. While for this down-sampled dataset few genes were significant after multiple testing correction, analysis of full datasets with larger sample sizes show statistically significant enrichment in most presented genes after multiple testing correction (data not shown).

### Regression of SBS9 and genomic features

The hdp per-Mb memory B cell mutational signature results above were used to identify genomic features associated with the location of mutations attributable to a particular mutational signature. To achieve a finer-scale genomic resolution, each Mb bin was further divided up into 10-kb bins, and the proportion of each mutational signature in a Mb bin was used to calculate a signature attribution per 10-kb bin, based on the type and trinucleotide context of mutations in the 10-kb bin.

The number of mutations attributable to a particular mutational signature, per 10-kb window, was regressed on each of 36 genomic features (Supplementary Table 4). Noise was further removed from the replication timing data, using the GM12878 blood cell line data, and filtering the Wave Signal data by removing low sum signal (<95) regions, per Hansen et al.[53]. SBS9 was analysed separately from the SBSblood/SBS1 combined signature. The number of mutations per signature per bin was calculated as the sum of the per-nucleotide probabilities per signature within a given bin. For the analysis of a given signature, a bin was only included if the average contribution of that signature was greater than 50%. This step ameliorates the problem of artificially high numbers of mutations being ascribed to a bin due to the combination of a trivially small attribution but a high overall mutation rate. This can occur in high SHM or SBS9 regions. This left 26,151 bins for SBS9 and 25,202 bins for SBSblood, out of 91,343 bins with mutations and 279,094 bins genome-wide. We also included a random sample of zero-mutation bins to equal 10% of the total bins.

We performed lasso-penalized general additive model regressions of the number of mutations per bin with the value of the genomic features. We used the gamsel function in R (package gamsel), with the lambda estimated from a fivefold cross-validation of training data (two-thirds of the data). To estimate individual effect sizes, we performed general additive model regressions per genomic feature using the function *gam* (R package *mgcv*). The same analysis was also performed on HSPC mutations. The results for the full and individual regression models for each of SBS9 and SBSblood/1 in memory B cells and for all HSPC mutations can be found in Supplementary Table 4.

### RAG and CSR motif analysis

We assessed the enrichment of V(D)J recombination (mediated by RAG) and class switch recombination (CSR, mediated by AID) associated motifs in regions proximal to lymphocyte structural variants. We identified the presence of full length and heptamer RSS motifs associated with RAG binding and endonuclease activity (RAG motifs) for the 50 bp flanking each structural variant breakpoint using the program FIMO[54] ($P < 10^{-4}$). Clusters of AGCT and TGCA repeats, associated with AID cytosine deamination and CSR (CSR motifs), were identified in the 1,000 bp flanking each structural variant breakpoint using the program MCAST[55] ($P < 0.1$, maximum gap = 100, $E < 10,000$). In order to estimate a genomic background rate of these motifs, we generated 100 genomic controls sets, randomly selected from regions of the genome not excluded from variant calling, and performed both the RAG and CSR motif analyses on these sets. The genomic background rate presented is the median of the 100 control datasets for each motif analysis. Both the RAG and CSR motif analyses were also performed for structural variants from the PCAWG cancer genomes included in the mutational signatures analysis and for acute lymphoblastic leukaemia genomes[3].

### Telomere length

We estimated the telomere length for HSPC and lymphocyte genomes (Supplementary Table 3) using the program Telomerecat[56]. Telomere lengths for all genomes for a given donor were estimated as a group.

### Timing of mutational processes

Following a procedure described previously[33,57], we modelled the distribution of somatic mutations along the genome from the density of chromatin immunoprecipitation–sequencing reads using random forest regression in a tenfold cross-validation setting and the LogCosh distance between observed and predicted profiles. Each mutation was attributed to the signature that most likely generated it and aggregated into 2,128 windows of 1 Mb spanning ~2.1 Gb of DNA. Signatures with an average number of mutations per window <1 were not evaluated due to lack of power. We determined the difference between models using a paired two-sided Wilcoxon test on the values from the tenfold cross-validation. Epigenetic data were gathered from different sources[58–60] (Supplementary Table 9) and consisted of 149 epigenomes representing 48 distinct blood cell types and differentiation stages and their replicates. Histone marks used included H3K27me3, H3K36me3, H3K4me1 and H3K9me3. To evaluate the specificity of SBS9 mutational profiles in memory B cells, we took the same number of mutations as in SBSblood with the highest association with SBS9 and compared models with an unpaired two-sided Wilcoxon test.

### Reporting summary

Further information on research design is available in the Nature Research Reporting Summary linked to this article.

### Data availability

Raw sequencing data are available at the European Genome–Phenome Archive (accession number EGAD00001008107). All somatic mutation calls and other relevant intermediate datasets are available on the github repository at https://github.com/machadoheather/lymphocyte_somatic_mutation.

### Code availability

An exhaustive repository of code for statistical analyses reported in this manuscript is available at https://github.com/machadoheather/lymphocyte_somatic_mutation.

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

**Acknowledgements** This work was supported by the WBH Foundation and Wellcome Trust. Investigators at the Sanger Institute are supported by a core grant from the Wellcome Trust. M.S.S. was the recipient of a Biotechnology and Biological Sciences Research Council Industrial Collaborative Awards in Science and Engineering PhD Studentship. The D.G.K. laboratory is supported by a Blood Cancer UK Bennett Fellowship (15008), an ERC Starting Grant (ERC-2016-STG–715371), a CR-UK Programme Foundation award (DCRPGF\100008) and an MRC-AMED joint award (MR/V005502/1). D.G.K., E.L. and A.R.G. are supported by a core support grant to the Wellcome MRC Cambridge Stem Cell Institute, Blood Cancer UK, the NIHR Cambridge Biomedical Research Centre, and the CRUK Cambridge Cancer Centre.

E.L. is supported by a Sir Henry Dale fellowship from Wellcome/Royal Society (107630/Z/15/Z), BBSRC (BB/P002293/1), and core support grants by Wellcome and MRC to the Wellcome-MRC Cambridge Stem Cell Institute (203151/Z/16/Z). K.K. and G.G. are supported by a GDAN grant (grant number U24CA210999). G.G. is partly supported by the Paul C. Zamecnik Chair in Oncology at the Massachusetts General Hospital Cancer Center. We thank F. Abascal, T. Coorens, T. Butler and S. Brunner for valuable guidance in data analysis; the CASM laboratory, including L. O'Neill and C. Latimer, for sample and data management, and CASM IT for technical support. This research was supported by the Cambridge NIHR BRC Cell Phenotyping Hub and staff, including E. Perez and N. Savinykh, who provided advice and support in flow cytometry and cell sorting. We are especially grateful to the tissue donors and their families and to the Cambridge Biorepository for Translational Medicine for the gift of tissue from transplant organ donors.

**Author contributions** H.E.M., P.J.C. and D.G.K. designed the experiments; P.J.C. and D.K. supervised the project; H.E.M. designed the lymphocyte expansion protocols with advice from D.G.K., D.J.H., N.F.Ø., M.B. and M.S.S; H.E.M. and M.D. performed the lymphocyte cell sorting and colony growth with advice from F.A.V.B., N.F.Ø., D.J.H., D.G.K. and E.L.; H.E.M. and M.D. performed the CellCelector colony picking with advice from C.M.; E.M. and N.F.Ø. performed the HSPC sorting and colony growth with advice from E.L.; H.E.M. performed the data analyses with advice from M.A.S., R.J.O., I.M., A.R.G., F.M. and P.J.C.; K.M. and K.S.-P. collected and processed samples; A.T.J.C. created the artwork for Fig. 1a; D.L. performed the CSR analysis; A.C. analysed the FACs data; K.K. performed the association of epigenetic marks and mutational signatures with advice from G.G. and P.P.; H.E.M. and P.J.C. wrote the manuscript; all authors reviewed and edited the manuscript.

**Competing interests** G.G. receives research funds from Pharmacyclics and IBM. G.G. is an inventor on multiple patents related to bioinformatics methods (MuTect, MutSig, ABSOLUTE, MSMutSig, MSMuTect, POLYSOLVER and TensorQTL). G.G. is a founder, consultant and holds privately held equity in Scorpion Therapeutics. D.J.H. receives research funding from AstraZeneca and D.G.K. receives research funding from STRM.bio. All other authors declare no competing interests.

**Additional information**
**Correspondence and requests for materials** should be addressed to David G. Kent or Peter J. Campbell.

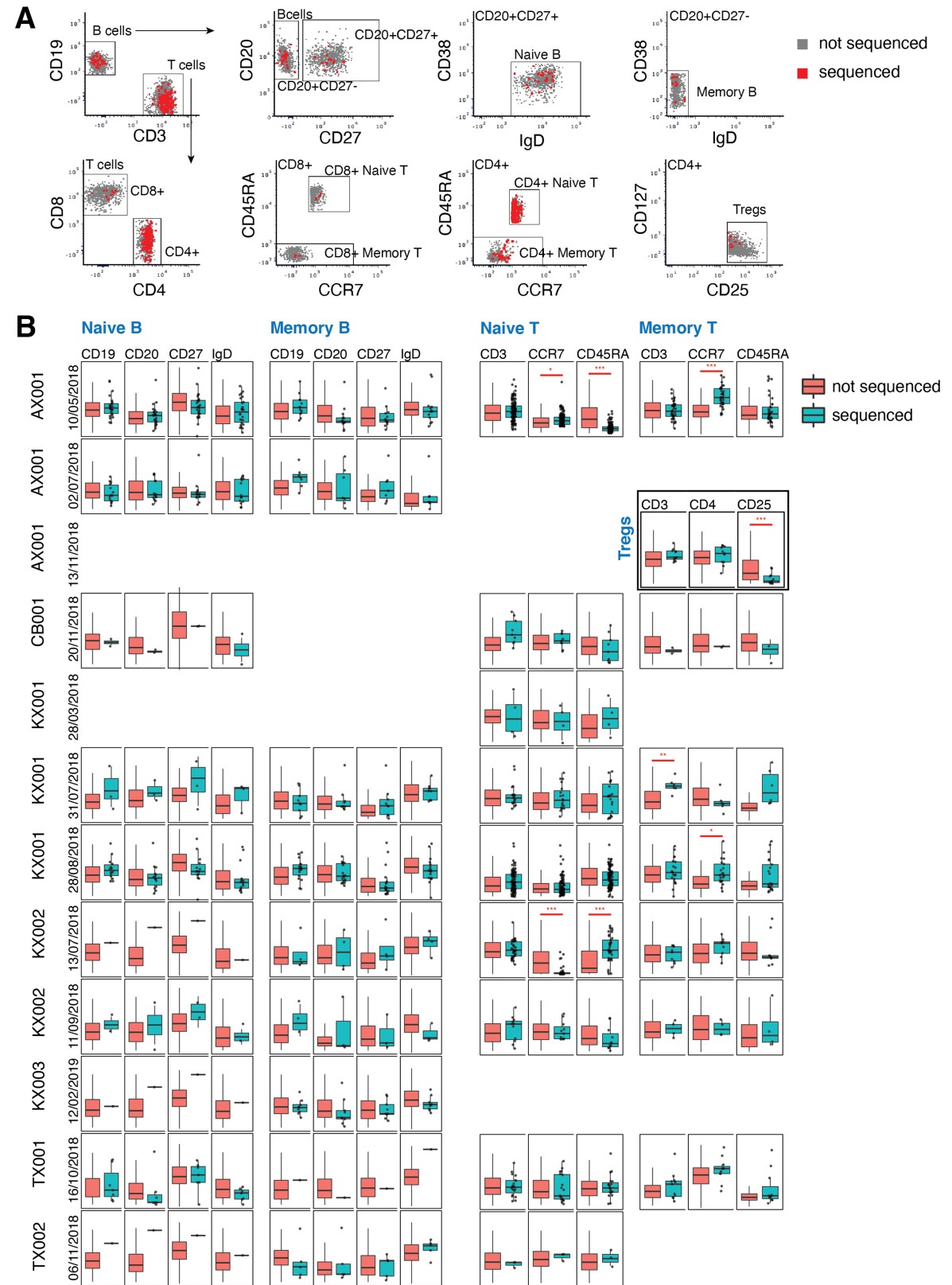

**Extended Data Fig. 1 | Assessment of culture bias by index flow-sorting.**
(A) Representative scatterplots of cell surface marker fluorescence intensity measured by flow cytometry (sort AX001 10/05/2018; AX001 13/11/2018 for Treg gate). Cells that successfully seeded colonies are coloured red; cells that did not form colonies are coloured grey. (B) Box-and-whisker plots showing fluorescence intensity for different cell surface markers in the various lymphocyte populations (columns) across different patients and days of flow-sorting (rows). Cells that successfully seeded colonies are shown in teal; cells that did not form colonies in orange. Boxes show the interquartile range and the centre horizontal lines show the median. Whiskers extend to the minimum of either the range or 1.5× the interquartile range. Red asterisks show a statistically significant difference between the fluorescence values of colony forming versus non-colony forming cells (two-sided *t*-test, false-discovery rate *q* < 0.05, **q* < 0.01, ***q* < 0.001, *P*-values in Table S10). The number of colony and non-colony forming cells per sort per subset can be found in Table S1.

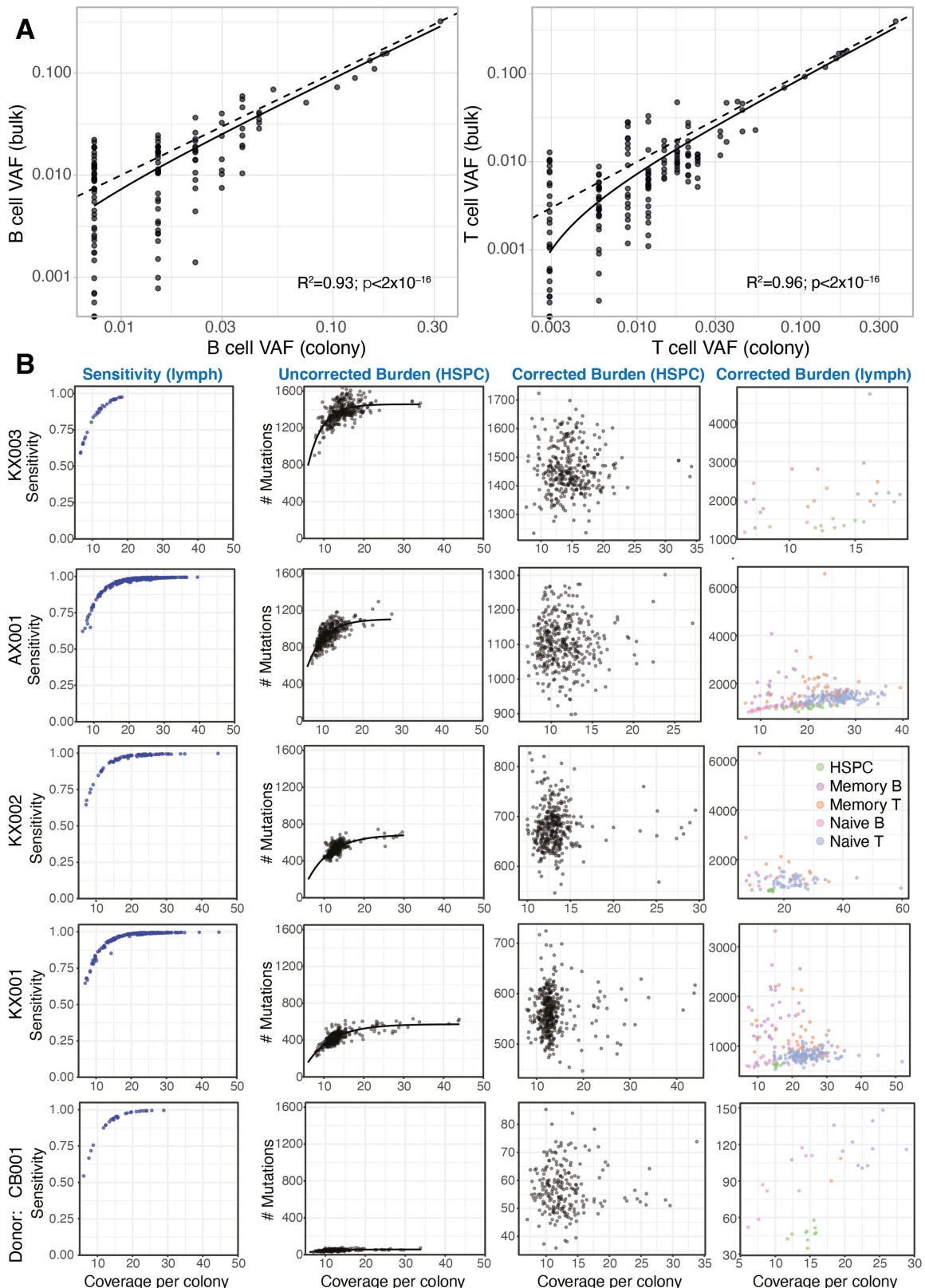

**Extended Data Fig. 2 | Clonal bias and sensitivity correction.** (A) To assess clone-to-clone biases in successfully seeded colonies, we reanalysed deep targeted resequencing data of bulk B and T cell lymphocytes from AX001[11]. The figure shows scatterplots of the fraction of lymphocyte colonies reporting a given somatic mutation (x-axis; log scale) with the variant allele fraction of that mutation in the bulk resequencing data (y-axis; log scale). Dashed lines are $x = y$ equality and solid lines show the linear regression fit (B cells, $R^2 = 0.47$, $P = 1 \times 10^{-18}$; T cells, $R^2 = 0.59$, $P = 2 \times 10^{-31}$). (B) Estimates of sensitivity for mutation calling as a function of depth for each colony (points in left panels) from each donor (rows; the 5 donors with the highest numbers of colonies are shown). The second column of panels shows uncorrected estimates of mutation burden for HSPCs in each donor, while the third column shows mutation burden estimates after correction for sequencing depth by asymptotic regression. The fourth column shows the corrected mutation burdens for lymphocyte colonies.

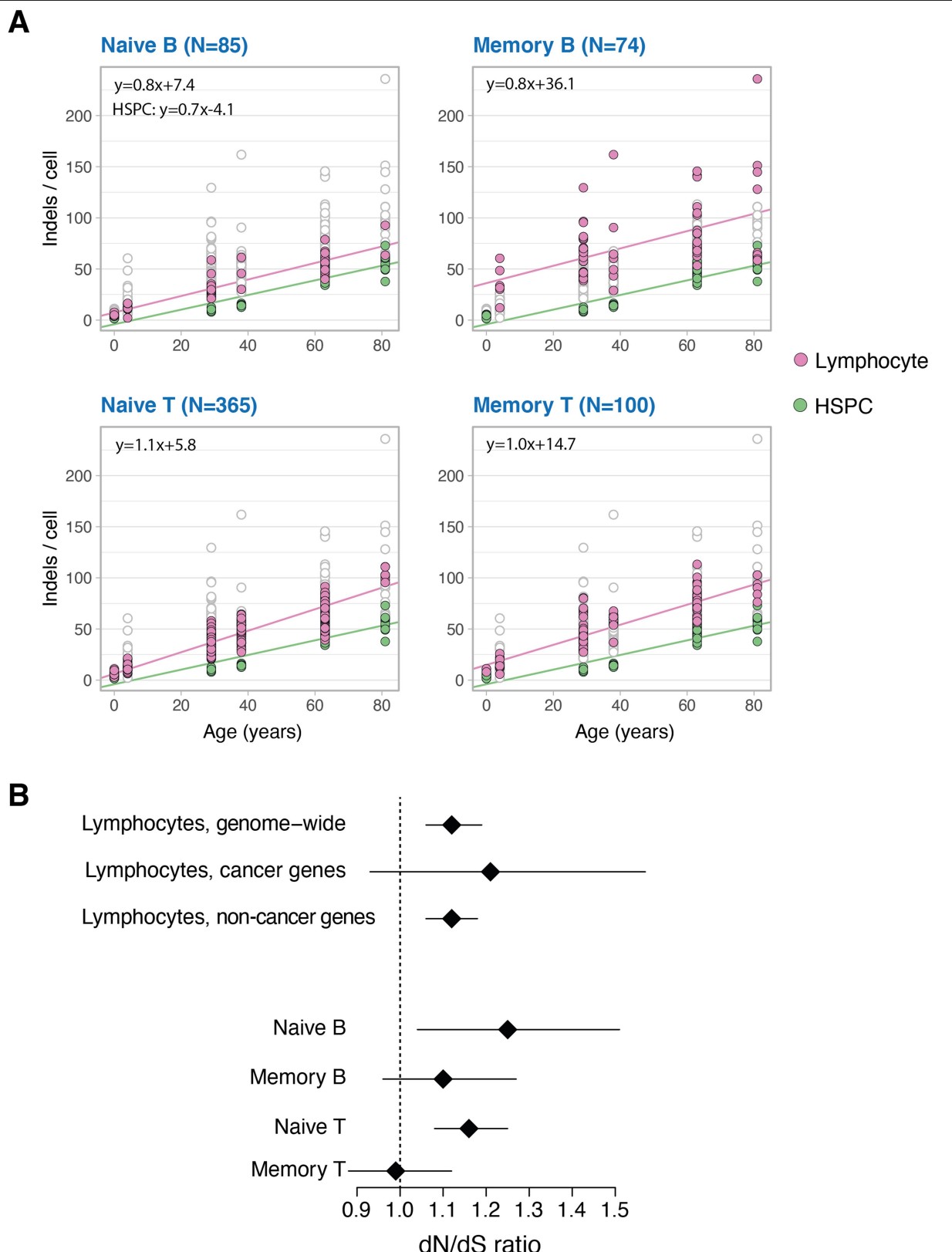

**Extended Data Fig. 3 | Indels and selection pressure.** (A) Indel mutation burden per genome for the four main lymphocyte subsets (pink points), compared with HSPCs (green points). Each panel has all genomes plotted underneath in white with grey outline. The lines show the fit by linear mixed effects models for the respective populations. (B) Plots of the estimated dN/dS ratio for mutations genome-wide (excluding immunoglobulin genes) for all lymphocytes, and for the various individual lymphocyte populations.

The second row shows the estimated dN/dS ratio for known cancer genes in all lymphocytes. The diamond shows the point estimates, and the lines the 95% confidence intervals. The point estimates / number of variants included in each analysis are as follows: lymphocytes, genome-wide = 1.12 / 7555; lymphocytes, cancer genes = 1.21 / 352; naive B = 1.25 / 671; memory B = 1.10 / 1132; naive T = 1.16 / 4162; memory T = 0.99 / 1414.

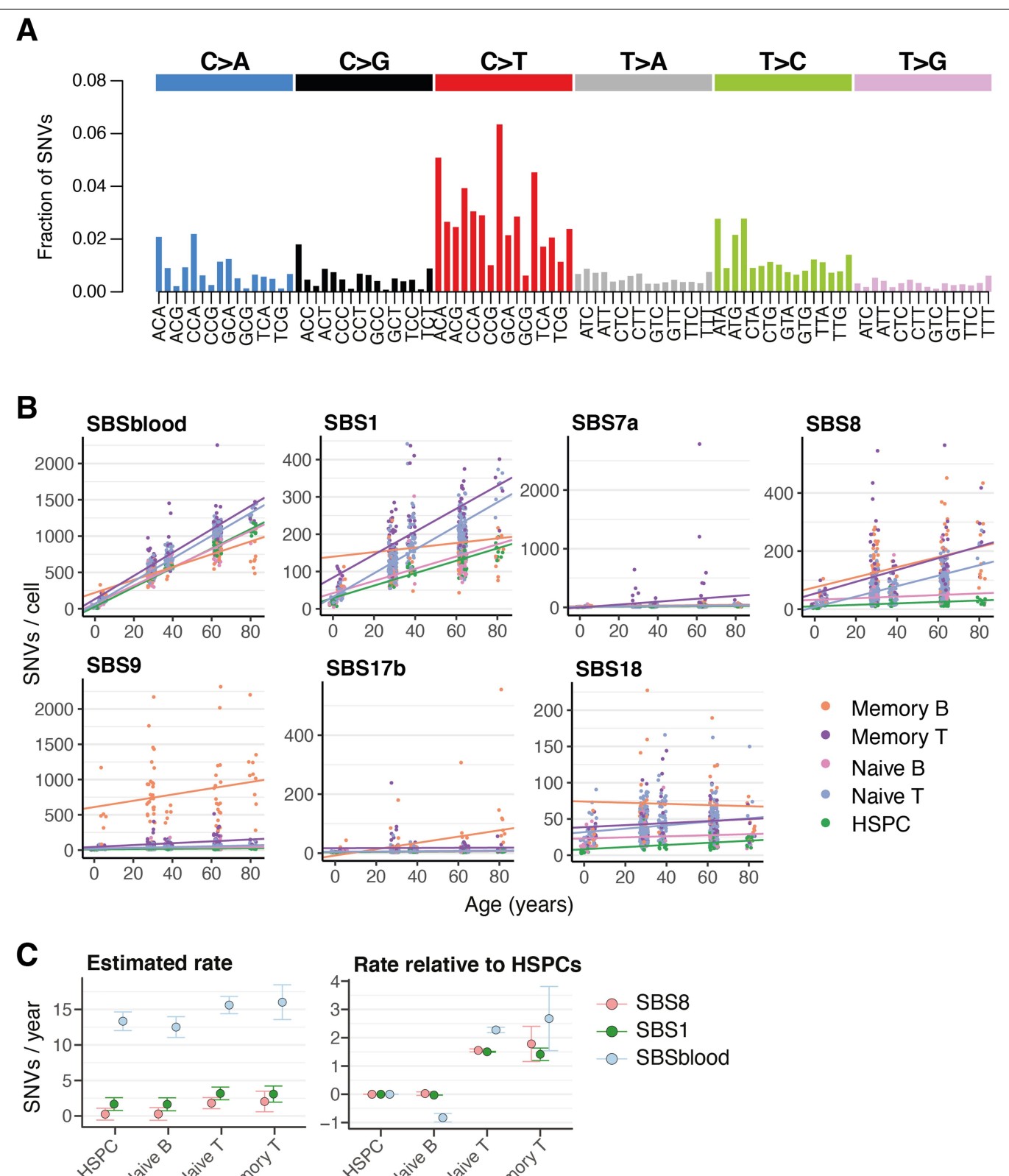

**Extended Data Fig. 4 | Mutational signatures by age.** (A) SBSblood signature identified using HSPC genomes and the program *sigfit*. Trinucleotide contexts on the *x*-axis represent 16 bars within each substitution class, divided into 4 sets of 4 bars, grouped by the nucleotide 5′ to the mutated base, and within each group by the 3′ nucleotide. (B) SNV mutation burden per genome, shown separately for each mutational signature. The lines show the fit by linear mixed effects models for the respective populations. Two outlier cells (PD40667vu and PD40667rx) are excluded from plotting. (C) The rate of mutation accumulation per year (slopes in B) for signatures with strong age effects. Error bars represent the 95% confidence intervals on the slope from the linear mixed effects models.

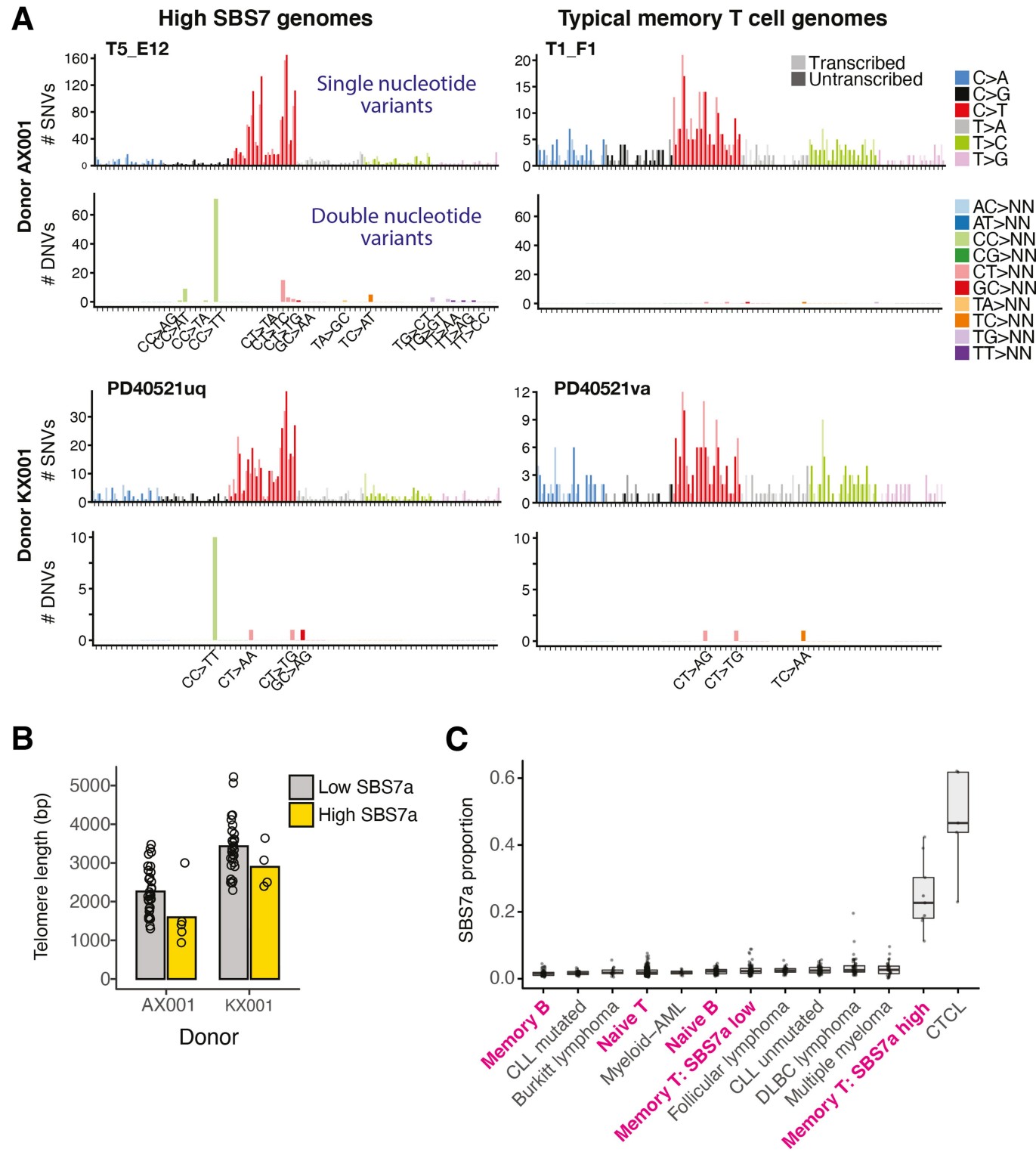

**Extended Data Fig. 5 | Ultraviolet light mutational signature (SBS7a) in lymphocytes.** (A) Raw mutational spectra shown for all mutation calls from four lymphocyte colonies, two with high contribution of SBS7a (left) and two with a more typical T-cell spectrum (right) from two different donors (rows). For each cell, the top panel shows the SNV spectrum, with trinucleotide contexts on the x-axis representing 16 bars within each substitution class, divided into 4 sets of 4 bars, grouped by the nucleotide 5′ to the mutated base, and within each group by the 3′ nucleotide. The bottom panel shows frequency of dinucleotide substitutions. (B) Telomere lengths for memory T cells with (yellow) and without (grey) high SBS7a signature. A memory T cell with high UV signature is defined as having greater than 9.5% (2 standard deviations above the mean) of its mutations attributable to SBS7a. (C) Proportion of mutations attributable to SBS7a across normal lymphocytes (paediatric samples excluded) and lymphoid malignancies. Boxes show the interquartile range and the centre horizontal lines show the median. Whiskers extend to the minimum of either the range or 1.5× the interquartile range. Number of genomes included per group: naive B: 68, memory B: 68, naive T: 332, memory T SBS7a low: 78, memory T SBS7a high: 9, Burkitt lymphoma: 17, CLL (chronic lymphocytic leukaemia) mutated: 38, CLL unmutated: 45, C. (cutaneous) T-cell lymphoma: 5, DLBC (Diffuse Large B-cell) lymphoma: 47, follicular lymphoma: 36, multiple myeloma: 30, myeloid-AML (acute myeloid leukaemia): 10.

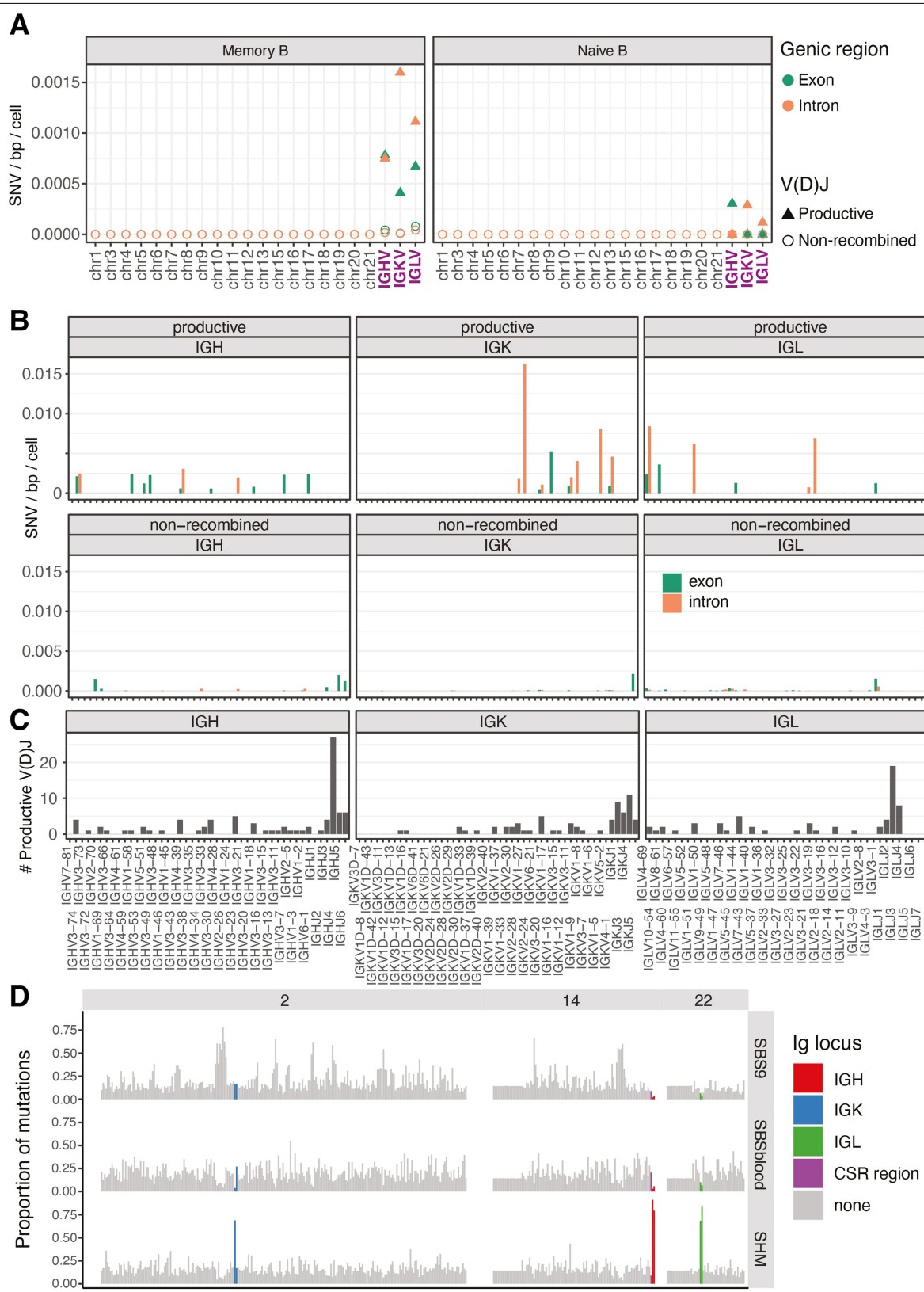

**Extended Data Fig. 6 | Distribution of mutational signatures across the genome.** (A) Estimates of the mutation rate across non-Ig chromosomes and Ig regions for memory (left) and naive B (right) cells. Rates for the Ig regions are calculated separately for the productive (triangles) and non-recombined alleles (circles) and exons (green) versus introns (orange). (B) Estimated mutation rates across different variable segments of the Ig genes for exons (green) versus introns (orange). (C) Number of productive V(D)J rearrangements affecting each variable segment in the dataset. (D) Proportion of mutations across chromosomes 2, 14 and 22 in each 1Mb window attributed to signatures SBS9, SBSblood and the canonical somatic hypermutation (SHM) signature (rows). Windows spanning the relevant immunoglobulin regions are coloured according to the key.

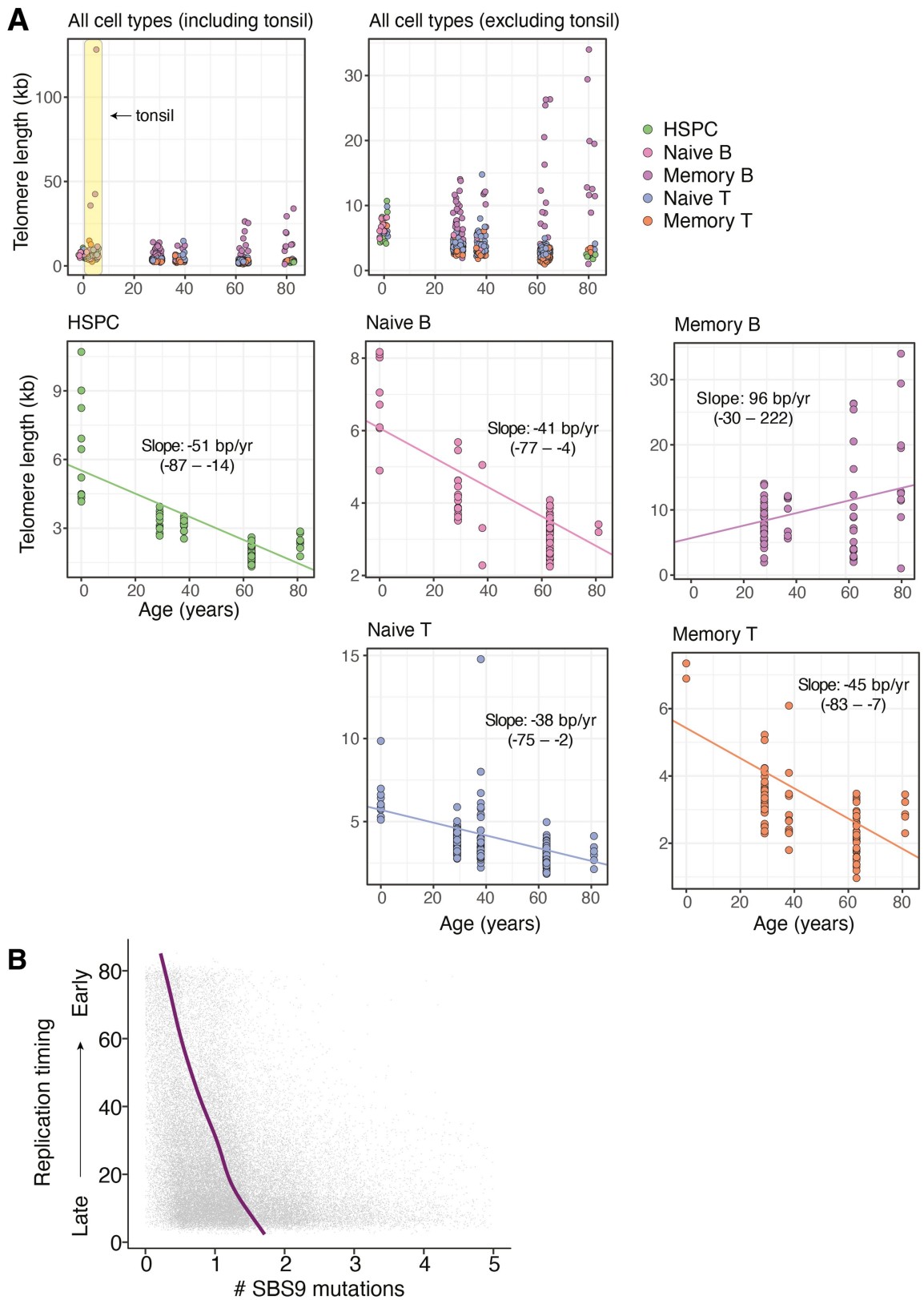

**Extended Data Fig. 7 | Telomere lengths and SBS9 versus replication timing.** (A) The top left panel includes the tonsil-derived genomes, which have an exceptionally high variance in telomere length. The remaining panels exclude these genomes, and show the estimated telomere lengths (*y*-axis) for each cell as a function of age (*x*-axis). Lines show the estimated fit by linear mixed effects models for each cell type, with the slope and 95% confidence intervals quoted in text. (B) Replication timing and number of SBS9 mutations per 10kb window. The line represents the GAM regression prediction. The x-axis is truncated at 5, excluding 0.3% of the data, and points have random noise (−0.5 to 0.5) to facilitate visualization.

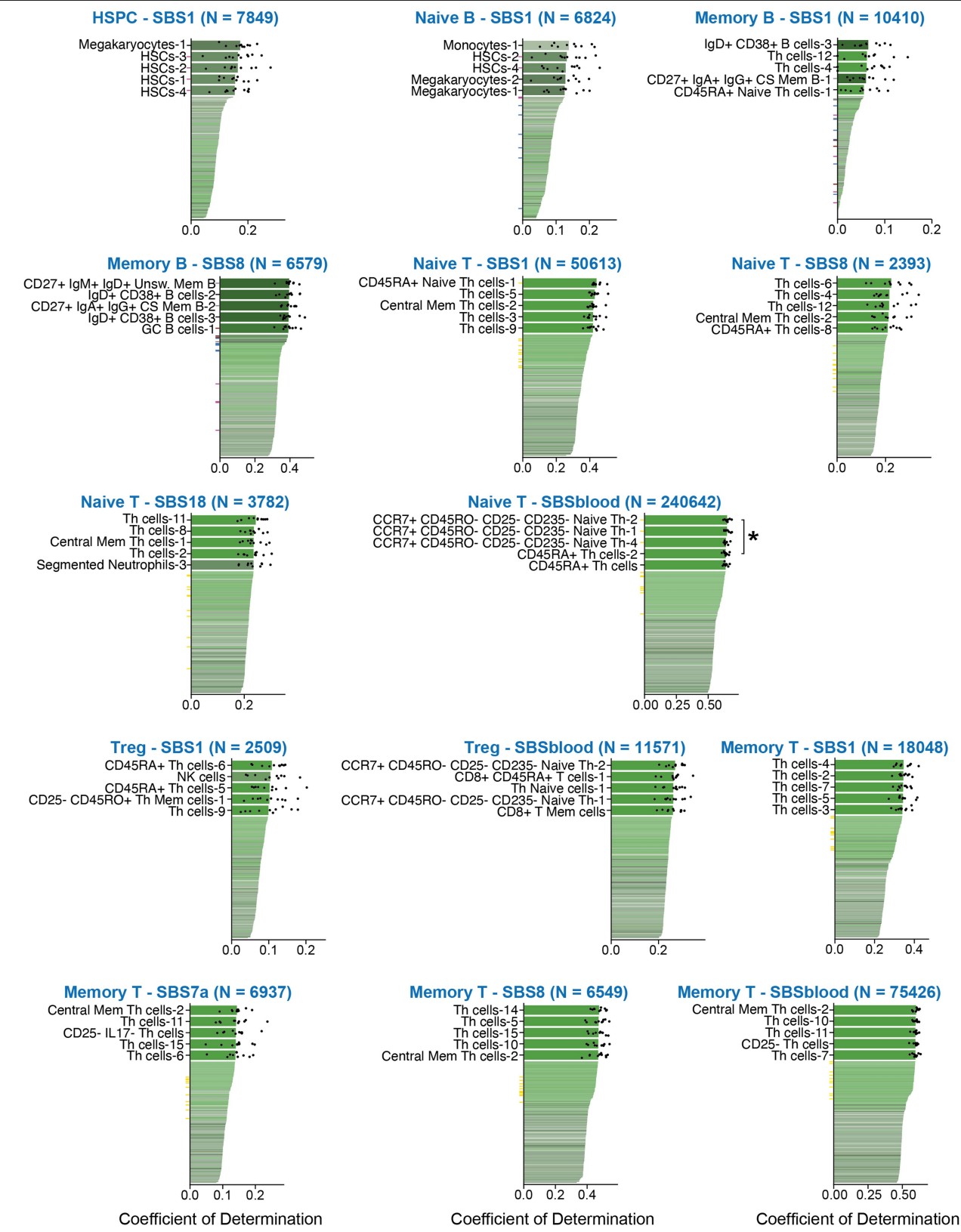

**Extended Data Fig. 8 | Relationships of signatures to epigenetic marks across haematopoietic cell types.** Performance of prediction of genome-wide mutational profiles attributable to particular mutational signatures from histone marks of 149 epigenomes representing distinct blood cell types and different phases of development (subscripts indicate replicates); ticks are coloured according to the epigenetic cell type (purple, HSC; blue, naive B cell; grey, memory B cell; maroon, GC B cell); black points depict values from ten-fold cross validation; *P*-values were obtained for the comparison of the 10-fold cross validation values using the two-sided Wilcoxon test (CS, class switched; GC, germinal centre; HSC, hematopoietic stem cell; Mem, memory).

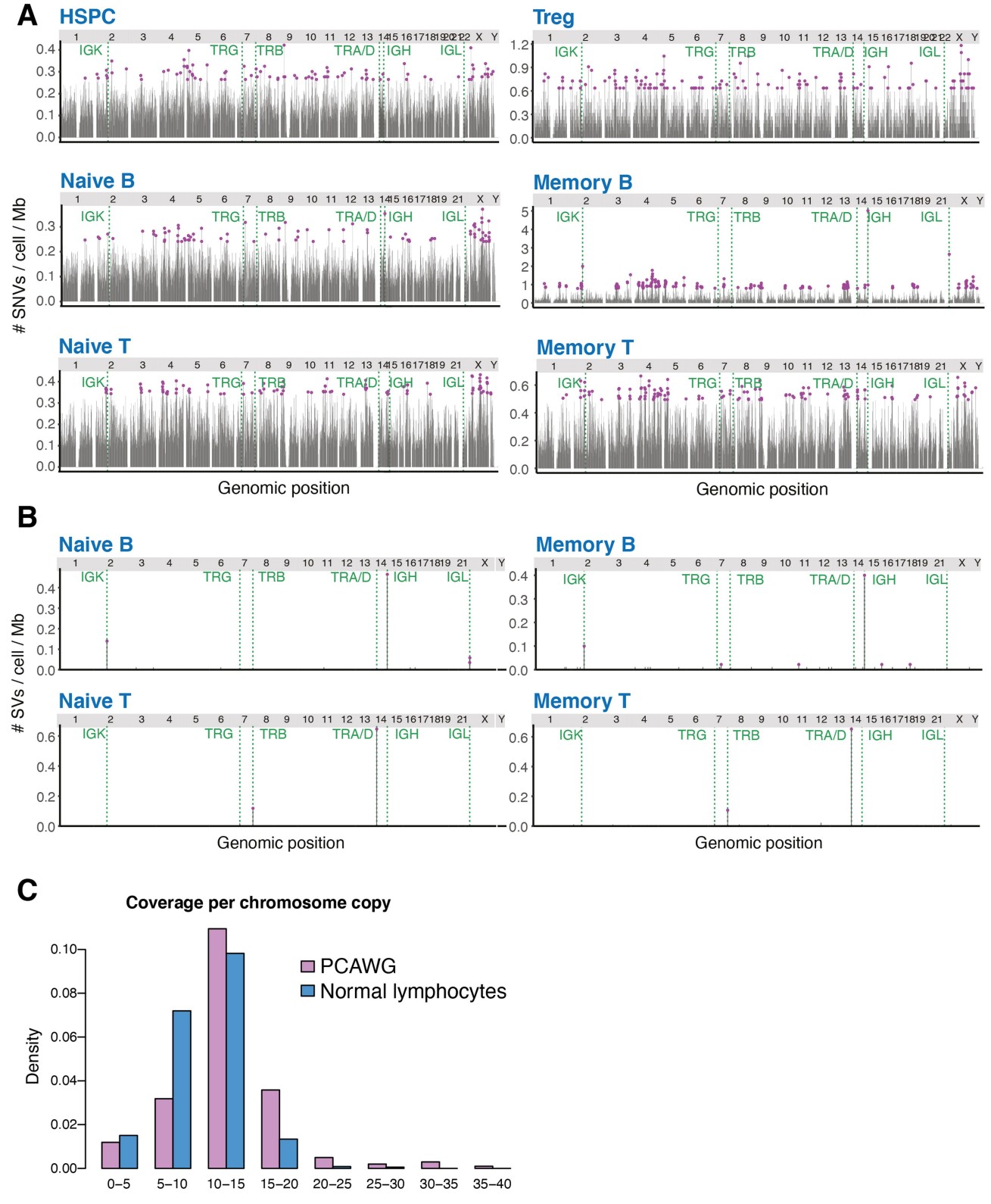

**Extended Data Fig. 9 | SV density and patterns in normal and malignant lymphocytes.** (A-B) Mutation rates per 1Mb bin across the genome for SNVs (A) and structural variants (B) split by cell type, with chromosomes labelled in the top strip, and Ig/TCR regions marked. Circles (purple) denote bins with more mutations than 2 standard deviations above the mean. (C) Histogram showing the distribution of estimated number of reads per informative chromosome copy for the normal lymphocytes (blue) and lymphoid malignancies from PCAWG (purple). For cancer genomes, purity and ploidy were estimated from the copy number patterns; for lymphocyte colonies, the purity was 1 and ploidy was 2.

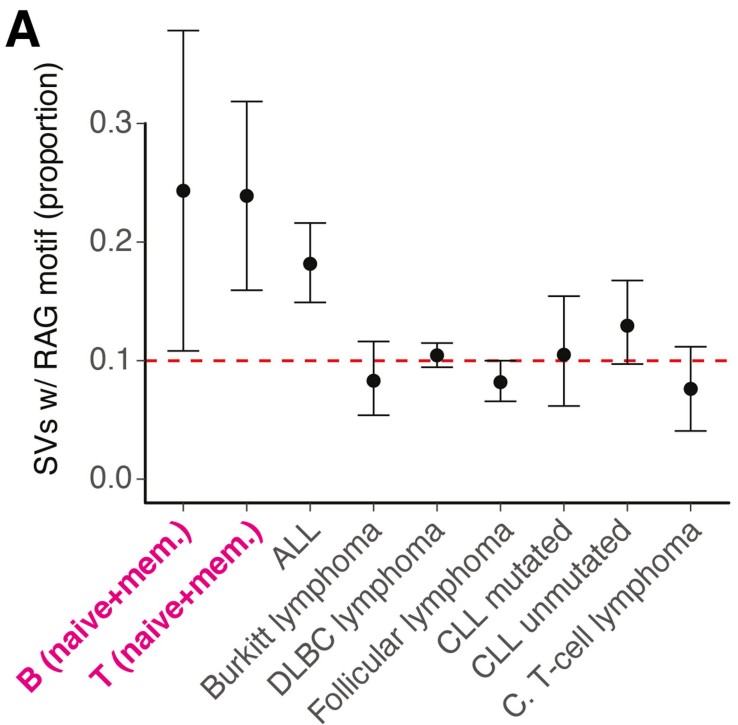

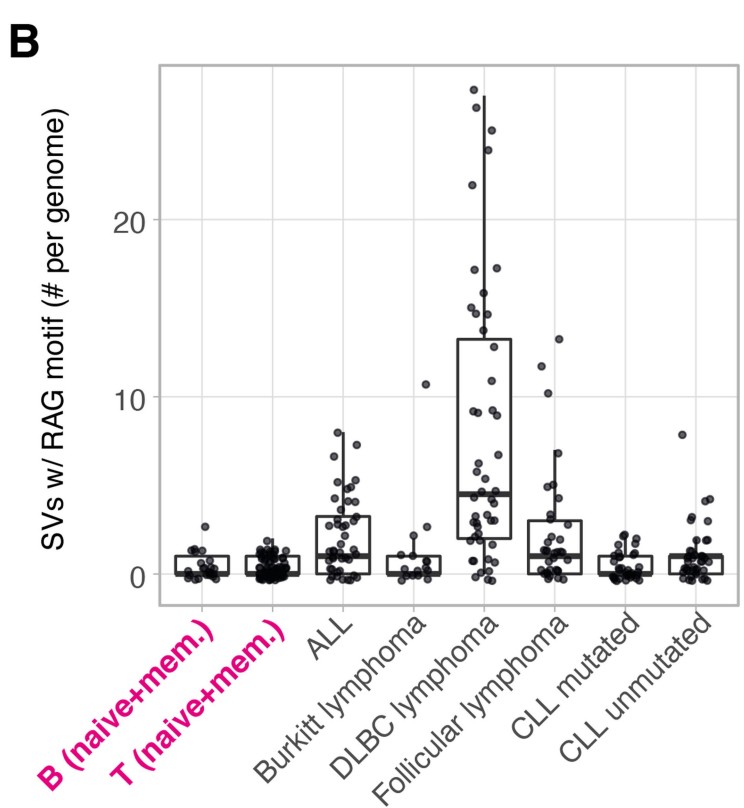

**Extended Data Fig. 10 | RAG-mediated SVs in normal versus malignant lymphocytes.** (A) Point estimates and 95% confidence intervals for the proportion of SVs with RSS (RAG) motifs within 50bp of a breakpoint. (B) Number of SVs with RSS (RAG) motifs within 50bp of a breakpoint. Boxes show the interquartile range and the centre horizontal lines show the median.

Whiskers extend to the minimum of either the range or 1.5× the interquartile range. Paediatric samples were excluded. Number of SVs per group: B = 145, T = 841, ALL = 523, Burkitt lymphoma = 305, CLL mutated = 252, CLL unmutated = 440, C. T-cell lymphoma = 204, DLBC lymphoma = 3754, follicular lymphoma = 1095.

David Kent

# Reporting Summary

## Statistics

For all statistical analyses, confirm that the following items are present in the figure legend, table legend, main text, or Methods section.

| n/a | Confirmed | |
|---|---|---|
| ☐ | ☒ | The exact sample size (*n*) for each experimental group/condition, given as a discrete number and unit of measurement |
| ☐ | ☒ | A statement on whether measurements were taken from distinct samples or whether the same sample was measured repeatedly |
| ☐ | ☒ | The statistical test(s) used AND whether they are one- or two-sided *Only common tests should be described solely by name; describe more complex techniques in the Methods section.* |
| ☐ | ☒ | A description of all covariates tested |
| ☐ | ☒ | A description of any assumptions or corrections, such as tests of normality and adjustment for multiple comparisons |
| ☐ | ☒ | A full description of the statistical parameters including central tendency (e.g. means) or other basic estimates (e.g. regression coefficient) AND variation (e.g. standard deviation) or associated estimates of uncertainty (e.g. confidence intervals) |
| ☐ | ☒ | For null hypothesis testing, the test statistic (e.g. *F*, *t*, *r*) with confidence intervals, effect sizes, degrees of freedom and *P* value noted *Give P values as exact values whenever suitable.* |
| ☒ | ☐ | For Bayesian analysis, information on the choice of priors and Markov chain Monte Carlo settings |
| ☐ | ☒ | For hierarchical and complex designs, identification of the appropriate level for tests and full reporting of outcomes |
| ☐ | ☒ | Estimates of effect sizes (e.g. Cohen's *d*, Pearson's *r*), indicating how they were calculated |

*Our web collection on statistics for biologists contains articles on many of the points above.*

## Software and code

Policy information about availability of computer code

| Data collection | No software was used for data collection. |
|---|---|
| Data analysis | Open source programs used (also stated in manuscript): BWA v0.7.17, CaVEMan v1.11.2, Pindel v3.1.2, BRASS v6.1.2, ASCAT 4.0.1, R v4.0.2, AnnotateBRASS v3, SigProfiler 1.0.0, IgCaller v1.2, FIMO v5.4.1, MCAST v5.4.1, Telomerecat v4.0.2, FlowJo v10.7.1, FCS Express v7.10.0007, Rstudio v1.4.1073. R packages used: hdp v0.1.5, sigfit v2.2., Rsamtools v2.2.3, MASS v7.3.57, GenomicRanges v1.38.0, plyr v1.8.5, ggplot2 v3.2.1, foreach v1.4.8, doParallel v1.0.16, reshape2 v1.4.3, sigfit v2.2, stringr v1.4.0, dplyr v0.8.4, RColorBrewer v1.1.2, BSgenome.Hsapiens.UCSC.hg19 v1.4.3, selectiveInference v1.2.5, gamsel v1.8.1, mgcv v1.8.31, grid v3.5.1, gridExtra v2.3, ggpubr v0.2.4, tidymv v3.2.0, GenomicFeatures v1.42.1, nrmisc v, tidyverse v1.3.0, magrittr v1.5, rtracklayer v1.50.0, BSgenome.Hsapiens.1000genomes.hs37d5 v0.99.1, cowplot v1.0.0 Custom code made available (also stated in manuscript): https://github.com/machadoheather/lymphocyte_somatic_mutation No commercial software used. |

For manuscripts utilizing custom algorithms or software that are central to the research but not yet described in published literature, software must be made available to editors and reviewers. We strongly encourage code deposition in a community repository (e.g. GitHub). See the Nature Portfolio guidelines for submitting code & software for further information.

## Data

Policy information about availability of data

All manuscripts must include a data availability statement. This statement should provide the following information, where applicable:

- Accession codes, unique identifiers, or web links for publicly available datasets
- A description of any restrictions on data availability
- For clinical datasets or third party data, please ensure that the statement adheres to our policy

Sequence data that support the findings of this study have been deposited in the European Genome-Phenome Archive (https://www.ebi.ac.uk/ega/home), accession EGAD00001008107. The 149 epigenetic datasets are from the ENCODE and IHEC studies and are described in Table S9. The genomic feature datasets are as follows (and described in Table S4): FeatureID: Data_source
ALU_rep_dist_log10: doi.org/10.1038/s41586-019-1913-9
centromere_dist_log10: doi.org/10.1038/s41586-019-1913-9
cpg_islands_dist_log10: doi.org/10.1038/s41586-019-1913-9
direct_rep_dist_log10: doi.org/10.1038/s41586-019-1913-9
DNA_rep_dist_log10: doi.org/10.1038/s41586-019-1913-9
DNAMethylSBS: doi.org/10.1038/s41586-019-1913-9
g4_dist_log10: doi.org/10.1038/s41586-019-1913-9
gc_content_value: doi.org/10.1038/s41586-019-1913-9
gene_dens_1e6: doi.org/10.1038/s41586-019-1913-9
H2A.Z: doi.org/10.1038/s41586-019-1913-9
H3K27me3: doi.org/10.1038/s41586-019-1913-9
H3K36me3: doi.org/10.1038/s41586-019-1913-9
H3K4me1: doi.org/10.1038/s41586-019-1913-9
H3K4me2: doi.org/10.1038/s41586-019-1913-9
H3K4me3: doi.org/10.1038/s41586-019-1913-9
H3K79me2: doi.org/10.1038/s41586-019-1913-9
H3K9ac: doi.org/10.1038/s41586-019-1913-9
H3K9me3: doi.org/10.1038/s41586-019-1913-9
H4K20me1: doi.org/10.1038/s41586-019-1913-9
L1_rep_dist_log10: doi.org/10.1038/s41586-019-1913-9
L2_rep_dist_log10: doi.org/10.1038/s41586-019-1913-9
LAD_dens_1e6: doi.org/10.1038/s41586-019-1913-9
LTR_rep_dist_log10: doi.org/10.1038/s41586-019-1913-9
MIR_rep_dist_log10: doi.org/10.1038/s41586-019-1913-9
recomb_rate_nearest_value: doi.org/10.1038/s41586-019-1913-9
rep_timing_Gm: http://genome.ucsc.edu/cgi-bin/hgFileUi?db=hg19&g=wgEncodeUwRepliSeq
RNAseq: doi.org/10.1038/s41586-019-1913-9
short_tandem_rep_dens_3e3: doi.org/10.1038/s41586-019-1913-9
SIMPLE_REPEAT_rep_dist_log10: doi.org/10.1038/s41586-019-1913-9
TAD_b_dist_log10: doi.org/10.1038/s41586-019-1913-9
telomere_dist_log10: doi.org/10.1038/s41586-019-1913-9
triplex_mirror_rep_dist_log10: doi.org/10.1038/s41586-019-1913-9

# Field-specific reporting

Please select the one below that is the best fit for your research. If you are not sure, read the appropriate sections before making your selection.

☒ Life sciences  ☐ Behavioural & social sciences  ☐ Ecological, evolutionary & environmental sciences

For a reference copy of the document with all sections, see nature.com/documents/nr-reporting-summary-flat.pdf

# Life sciences study design

All studies must disclose on these points even when the disclosure is negative.

| | |
|---|---|
| Sample size | We optimised the number of individuals (7) and number of genomes per cell subset per individual (average of 102 genomes per individual) to describe the general mutational landscape per cell subset across a range of ages. No power calculation was performed, and there was no target effect size. The samples were chosen to have a broad distribution across ages, from birth (cord blood) up to 81 years of age, where we would expect to start seeing clonal haematopoiesis. Samples were spaced as evenly as possible across ages, with the limitation of pediatric samples, for which only 4 year old samples were obtainable. Previous studies had found an average mutation rate of 16 mutations per cell per year, which indicated that sampling 7 individuals along the described age range would allow for a statistically significant estimates of mutation rates in lymphocytes. |
| Data exclusions | Per pre-established criteria, genomes with a sequencing depth of less than 6x or and average VAF of less than 20% were excluded. This removed a total 39 genomes: 15 KX001 genomes, 12 KX002 genomes and 12 KX003 genomes. |

| | |
|---|---|
| Replication | While the specific samples used have been exhausted, most of the results from this study should be generally reproducible in separate healthy individuals of the same age, using the protocols and code included in this manuscript. |
| Randomization | This is not relevant to our study. All individuals were hematopoietically normal, and there was no test versus control groups. |
| Blinding | Blinding was not relevant to our study. The study only included samples of normal lymphocytes, and no tests were performed that required blinding. |

# Reporting for specific materials, systems and methods

We require information from authors about some types of materials, experimental systems and methods used in many studies. Here, indicate whether each material, system or method listed is relevant to your study. If you are not sure if a list item applies to your research, read the appropriate section before selecting a response.

## Materials & experimental systems

| n/a | Involved in the study |
|---|---|
| ☐ | ☒ Antibodies |
| ☒ | ☐ Eukaryotic cell lines |
| ☒ | ☐ Palaeontology and archaeology |
| ☒ | ☐ Animals and other organisms |
| ☐ | ☒ Human research participants |
| ☒ | ☐ Clinical data |
| ☒ | ☐ Dual use research of concern |

## Methods

| n/a | Involved in the study |
|---|---|
| ☒ | ☐ ChIP-seq |
| ☐ | ☒ Flow cytometry |
| ☒ | ☐ MRI-based neuroimaging |

## Antibodies

| | |
|---|---|
| Antibodies used | Antibody; Company; Clone; Catalogue Number; Flurophore; Dilution; Panel<br>CD3; BD; HIT3a; 555339; FITC; 1:500; HSPC_nonAX001<br>CD90; Biolgend; 5E10; 328110; PE; 1:50; HSPC_nonAX001<br>CD49f; BD; GoH3; 551129; PECy5; 1:100; HSPC_nonAX001<br>CD19; Biolgend; HIB19; 302226; A700; 1:300; HSPC_nonAX001<br>CD34; Biolgend; 581; 343514; APCCy7; 1:100; HSPC_nonAX001<br>Zombie ; Biolgend; NA; 423101; Aqua; 1:2000; HSPC_nonAX001<br>CD38; Biolgend; HIT2; 303516; PECy7; 1:100; HSPC_nonAX001<br>CD45RA; Biolgend; HI100; 304130; BV421; 1:100; HSPC_nonAX001<br>CD38; BD; HIT2; 560982; FITC; 1:100; HSPC_AX001<br>CD135; Biolegend; BV10A4H2; 313306; PE; 1:100; HSPC_AX001<br>CD34; BD; 581; 560710; PE-Cy7; 1:100; HSPC_AX001<br>CD90; BD; 5E10; 561971; APC; 1:100; HSPC_AX001<br>CD10; Biolegend; HI10a; 312212; APC-Cy7; 1:100; HSPC_AX001<br>CD45RA; BD; HI100; 562298; V450; 1:100; HSPC_AX001<br>CD3; Tonbo Biosciences; Hit3a; 20-0039-T100; APC; 1:80; lymphocyte_nonTreg<br>CD4; Biolegend; OKT4; 317441; BV785; 1:80; lymphocyte_nonTreg<br>CD8; BD; RPA-T8; 563821; BV650; 1:40; lymphocyte_nonTreg<br>CD19; Biolegend; HIB19; 302226; AF700; 1:80; lymphocyte_nonTreg<br>CD20; Biolegend; 2H7; 302347; PE Dazzle; 1:80; lymphocyte_nonTreg<br>CD27; BD; M-T271; 562513; BV421; 1:80; lymphocyte_nonTreg<br>CD38; Biolegend; HIT2; 356610; FITC; 1:80; lymphocyte_nonTreg<br>CD45RA; Biolegend; HI100; 560362; PerCP Cy5.5; 1:80; lymphocyte_nonTreg<br>CCR7; Biolegend; G043H7; 353227; BV711; 1:80; lymphocyte_nonTreg<br>IgD; Biolegend; IA6-2; 348209; PeCy7; 1:100; lymphocyte_nonTreg<br>live; Biolegend; n/a; 423101; Zombie aqua; 1:400; lymphocyte_nonTreg<br>CD3; Tonbo Biosciences; Hit3a; 20-0039-T100; APC; 1:80; Treg<br>CD4; Biolegend; OKT4; 317441; BV785; 1:80; Treg<br>CD8; BD; RPA-T8; 563821; BV650; 1:40; Treg<br>CD19; Biolegend; HIB19; 302226; AF700; 1:80; Treg<br>CD45RA; Biolegend; HI100; 560362; PerCP Cy5.5; 1:80; Treg<br>CD56; Biolegend; 39D5; 355503; PE; 1:80; Treg<br>CCR7; Biolegend; G043H7; 353227; FITC; 1:80; Treg<br>CD25; Biolegend; BC96; 302607; PeCy5; 1:80; Treg<br>CD127; Biolegend; A019D5; 351319; PeCy7; 1:80; Treg<br>CD69; Biolegend; FN50; 310921; AF700; 1:80; Treg<br>CD103; Biolegend; Ber-ACT8; 350213; BV421; 1:80; Treg<br>CCR9; Biolegend; L053E8; 358903; PE; 1:80; Treg<br>live; Biolegend; n/a; 423101; Zombie aqua; 1:400; Treg |
| Validation | These were all previously validated commercially available antibodies. Manufacturer validation and references for each can be found:<br>Antibody; Flurophore; Company; Catalogue Number; ManufacturerInformation<br>CCR7; FITC; Biolegend; 353227; https://www.biolegend.com/fr-ch/products/fitc-anti-human-cd197-ccr7-antibody-7537 |

CCR9; PE; Biolegend; 358903; https://www.biolegend.com/de-de/products/pe-anti-human-cd199-ccr9-antibody-8761

CD10; APC-Cy7; Biolegend; 312212; https://www.biolegend.com/fr-ch/products/apc-cyanine7-anti-human-cd10-antibody-4034

CD103; BV421; Biolegend; 350213; https://www.biolegend.com/de-de/products/brilliant-violet-421-anti-human-cd103-integrin-alphae-antibody-9746

CD127; PeCy7; Biolegend; 351319; https://www.biolegend.com/fr-fr/products/pe-cyanine7-anti-human-cd127-il-7ralpha-antibody-7216

CD135; PE; Biolegend; 313306; https://www.biolegend.com/fr-ch/products/pe-anti-human-cd135-flt-3-flk-2-antibody-2359

CD19; AF700; Biolegend; 302226; https://www.biolegend.com/it-it/products/alexa-fluor-700-anti-human-cd19-antibody-3399

CD20; PE Dazzle; Biolegend; 302347; https://www.biolegend.com/de-de/products/pe-dazzle-594-anti-human-cd20-antibody-10436

CD25; PeCy5; Biolegend; 302607; https://www.biolegend.com/en-gb/products/pe-cyanine5-anti-human-cd25-antibody-617

CD27; BV421; BD; 562513; https://www.bdbiosciences.com/en-us/products/reagents/flow-cytometry-reagents/research-reagents/single-color-antibodies-ruo/bv421-mouse-anti-human-cd27.562513

CD3; FITC; BD; 555339; https://www.bdbiosciences.com/en-us/products/reagents/flow-cytometry-reagents/research-reagents/single-color-antibodies-ruo/fitc-mouse-anti-human-cd3.555339

CD3; APC; Tonbo Biosciences; 20-0039-T100; https://tonbobio.com/products/apc-anti-human-cd3-hit3a

CD34; APCCy7; Biolgend; 343514; https://www.biolegend.com/fr-lu/products/apc-cyanine7-anti-human-cd34-antibody-6159

CD34; PE-Cy7; BD; 560710; https://www.bdbiosciences.com/en-gb/products/reagents/flow-cytometry-reagents/research-reagents/single-color-antibodies-ruo/pe-cy-7-mouse-anti-human-cd34.560710

CD38; PECy7; Biolgend; 303516; https://www.biolegend.com/fr-ch/products/pe-cyanine7-anti-human-cd38-antibody-5418

CD38; FITC; Biolegend; 356610; https://www.biolegend.com/en-ie/products/fitc-anti-human-cd38-antibody-14047

CD38; FITC; BD; 560982; https://www.bdbiosciences.com/en-us/products/reagents/flow-cytometry-reagents/research-reagents/single-color-antibodies-ruo/fitc-mouse-anti-human-cd38.560982

CD4; BV785; Biolegend; 317441; https://www.biolegend.com/fr-lu/products/brilliant-violet-785-anti-human-cd4-antibody-7978

CD45RA; BV421; Biolgend; 304130; https://www.biolegend.com/de-de/products/brilliant-violet-421-anti-human-cd45ra-antibody-7200

CD45RA; PerCP Cy5.5; Biolegend; 560362; https://www.biolegend.com/en-us/search-results/percp-cyanine5-5-anti-human-cd45ra-antibody-4241

CD45RA; V450; BD; 562298; https://www.bdbiosciences.com/en-gb/products/reagents/flow-cytometry-reagents/research-reagents/single-color-antibodies-ruo/v450-mouse-anti-human-cd45ra.560363

CD49f; PECy5; BD; 551129; https://www.bdbiosciences.com/zh-cn/products/reagents/flow-cytometry-reagents/research-reagents/single-color-antibodies-ruo/pe-cy-5-rat-anti-human-cd49f.551129

CD56; PE; Biolegend; 355503; https://www.biolegend.com/en-us/products/pe-anti-human-cd56-subset-msc-marker-antibody-8191

CD69; AF700; Biolegend; 310921; https://www.biolegend.com/fr-ch/products/alexa-fluor-700-anti-human-cd69-antibody-3425

CD8; BV650; BD; 563821; https://www.bdbiosciences.com/ko-kr/products/reagents/flow-cytometry-reagents/research-reagents/single-color-antibodies-ruo/bv650-mouse-anti-human-cd8.563821

CD90; PE; Biolgend; 328110; https://www.biolegend.com/it-it/products/pe-anti-human-cd90-thy1-antibody-4114

CD90; APC; BD; 561971; https://www.bdbiosciences.com/en-ca/products/reagents/flow-cytometry-reagents/research-reagents/single-color-antibodies-ruo/fitc-mouse-anti-human-cd90.555595

IgD; PeCy7; Biolegend; 348209; https://www.biolegend.com/en-us/search-results/pe-cyanine7-anti-human-igd-antibody-6996

live; Zombie aqua; Biolegend; 423101; https://www.biolegend.com/fr-fr/products/zombie-aqua-fixable-viability-kit-8444

# Human research participants

Policy information about studies involving human research participants

| Population characteristics | Describe the covariate-relevant population characteristics of the human research participants (e.g. age, gender, genotypic information, past and current diagnosis and treatment categories). If you filled out the behavioural & social sciences study design questions and have nothing to add here, write "See above." |
| Recruitment | Describe how participants were recruited. Outline any potential self-selection bias or other biases that may be present and how these are likely to impact results. |
| Ethics oversight | Identify the organization(s) that approved the study protocol. |

Note that full information on the approval of the study protocol must also be provided in the manuscript.

# Flow Cytometry

## Plots

Confirm that:

☒ The axis labels state the marker and fluorochrome used (e.g. CD4-FITC).

☒ The axis scales are clearly visible. Include numbers along axes only for bottom left plot of group (a 'group' is an analysis of identical markers).

☒ All plots are contour plots with outliers or pseudocolor plots.

☒ A numerical value for number of cells or percentage (with statistics) is provided.

## Methodology

| Sample preparation | All samples were received as viably frozen human blood mononuclear cells (MNCs) obtained from bone marrow, spleen, tonsil and peripheral blood from seven individuals. |

| | |
|---|---|
| Instrument | Sorting was performed on FACSAria III or FACSAria Fusion (BD Biosciences). |
| Software | During the sort data were collected using the FACSAria III or FACSAria Fusion (BD Biosciences) default software. Plotting of the gating strategy supplementary figure was done on FlowJo and plotting of the index sort data for the culture bias supplementary figure was done using FCS Express. |
| Cell population abundance | This study used no data on relative cell abundances for any of the samples. |
| Gating strategy | The FSC/SSC gate was set using the manual gating tool to exclude debris and small particles and to identify the lymphocyte fraction of cells as displayed in Supplementary Figure S1. Further gating was as follows: naive B lymphocytes (CD3-CD19 +CD20+CD27-CD38-IgD+), memory B lymphocytes (CD3-CD19+CD20+CD27+CD38-IgD-), naive T lymphocytes (CD3+CD4/CD8 +CCR7+CD45RAhigh), memory T lymphocytes (CD3+CD4/CD8+CD45RA-), regulatory T cells (Tregs: CD3+CD4 +CD25highCD127-) and HSPCs (CD3-CD19-CD34+CD38-CD90+CD45RA-). HSPCs from AX001 included HSCs (CD34+CD38-) and progenitors (CD34+CD38+CD10-/dim). |

☒ Tick this box to confirm that a figure exemplifying the gating strategy is provided in the Supplementary Information.

