## [Peer Review File · Nature]

Manuscript Title: Diverse mutational landscapes in human lymphocytes.

Reviewer Comments & Author Rebuttals

Reviewer Reports on the Initial Version:

Referees' comments:

Referee #1 (Remarks to the Author):

The manuscript by Machado et al. examines the accumulation of mutations in human lymphocytes across their development. To accomplish this, the research team performed whole genome sequencing on colonies derived from sorted naive and memory B and T lymphocytes. The authors have previously optimized protocols for expanding flow-sorted single lymphocytes and have convincingly demonstrated that majority of mutations detected in these colonies accurately represent genomic events in the sorted cells. The accumulation of mutations is evaluated as a function of age of the donor and stage of lymphocyte differentiation (naive vs. memory). The findings, that mutations accumulate as a function of lymphocyte diversification process and that lymphocytes have significantly more mutations than HSCs is fundamental to our understanding of the immune cell development and lymphomagenesis. Few groups are able to do WGS of individual lymphocytes without genome amplification and the researchers perform in depth analysis on the valuable data in hand from their sequencing efforts as well as prior sequencing of lymphoma cells to examine the contribution of mutations and off-target rearrangements to lymphomagenesis. I believe the presented work is of interest to the broader audience and is appropriate for the journal.

Computational methods are well described, manuscript reads well and is easy to understand. Conclusions are drawn based on sequence analysis of >70 colonies per cell type with appropriate statistical tests utilized. There are a few analysis/comparisons that are well within grasp given the data in hand that would further strengthen the manuscript.

For B cells:

- A lot of analysis that infers past GC experience is done on sorted memory B cells. Can the authors (1) confirm that majority of these lymphocytes have undergone CSR - not just via IgD-gating but sequence analysis of the constant region (2) evaluate SHM load in the intronic region of Ig rearrangement to compare mutation load at the Ig loci vs. other chromosomes? Figure 3B shows the increase in SBS9 signature in the recombined IgH region but it unclear precisely how this burden compares to other genomic regions, and what the spatial distribution of these mutations is. Supplementary figure S7 suggests that there is a significant increase on Ch14, presumably at IgH - but whether this is restricted to productive Ig rearrangements, is on both alleles and what the distribution is across Igh and Igl should be shown.
- It is not immediately clear why the authors expect 1-2 V(D)J events per lymphocyte, as each T or B cell would have experienced at least 2 (IgH and IgL; TCRA and TCRb) and possibly more if the initial recombination event is not successful. So this sentence seems a bit out of tune: "We found 1037 SVs across 635 lymphocytes, of which 85% occurred in Ig/TCR regions, consistent with 1-2 V(D)J recombination events per lymphocyte"?

For analysis of T cells:

- The authors demonstrate the B cell malignancies harbored a comparable number of mutations to "normal" B lymphocytes with a dominant SBS9 signature, but had elevated number of structural variation. The authors should examine T cell lymphomas (and in particular cutaneous T cell lymphomas) in light of the UV signature found in some "normal" memory T cells. WGS has previously be performed in CTCL and similar analysis can be done in data from CTCL vs. memory T

cells - to examine the nature and frequency of mutations and SVs.

For both T and B:

- Total mutation "burden" would be easier to appreciate if it is plotted across genome for each cell type with Ig and TCR alleles highlighted. This may make it easier for the reader to appreciate existence of any specific hotspots and relative frequency compared to mutations associated with VDJ rearrangement and SHM. Similar to what is done in figure S7 for the SBS9 signature
- Same for structural variation

General minor points:

- In discussing data in Fig S2 - please state the indel rate in B and T cells when comparing to HSC. Currently only the indel rate in HSCs is given.
- The statement pertaining to association of telomere length on pg. 8 appears to be a bit bold "These data confirm that telomeres do lengthen in the germinal centre reaction, and provide further evidence that off-target SBS9 mutations are generated in the germinal centre." as the analysis was done on memory B cells, and while many may have gone through the GC reaction the authors have not demonstrated that.

Lastly, presumably 20x coverage for somatic mutation was, at least in part, based on practical/financial considerations. The analysis is "benchmarked" against HSCs. However, given the long-standing convention that 30x coverage minimum is required for confident mutation calling, the authors should perhaps address that in a sentence within the text...

This paper compares SNV calling from 2x-100x and found 30x was the minimum for high confidence:

<https://www.ncbi.nlm.nih.gov/pmc/articles/PMC5591230/>

Illumina recommendations:

https://www.illumina.com/Documents/products/technotes/technote_snp_caller_sequencing.pdf

Referee #2 (Remarks to the Author):

In this study, Machado et al. set out to comprehensively determine the mutations rate in human T- and B- lymphocytes at various stages of development and in different tissues. They obtained tissue from 5 sources (marrow, cord blood, peripheral blood, spleen, and tonsil) and 8 subsets were sorted, and 6 cell types were grown into colonies used for WGS (naïve and memory T and B-cells, T reg, and HSPCs). Novel culturing methods were developed to allow for in vitro expansion. They find that mutation burden associates with increasing age and cell type, with memory cell types having a higher burden than naïve. Mutational processes were similar in naïve lymphocytes and HSPCs, but was greater variability in mutational burden in memory cells. This was attributable largely to the mutations introduced in the germinal center reaction in the IGHV locus (SBS9), but also additional novel signatures contributed (UV light and GI exposure). SBS9 was associated with late replicating, closed chromatin regions of the genome, so may be related to error-prone bypass by polymerase ϵ . Indels in lymphocytes were associated with RAG activity, but not class switching. They further find that increased mutational burden in some lymphoid malignancies is due to heightened mutation rate, but with the same processes as in normal lymphocytes.

The paper is well written and technically a tour-de-force on mutational processes in lymphocytes. The conclusions are mostly justified. Since the paper is entirely descriptive, some of the conclusions would be strengthened by experimental validation, for example the polymerase ϵ

bypass hypothesis for SBS9. The development of culturing methods for human lymphocytes seems to be a new advance, but requires much more detail so others can replicate. There should also be a supplemental method describing the results of the expansion for each cell type from each donor. How efficient was the expansion? Did a large proportion of cells not form colonies? If so, this could bias the results of the sequencing and influence the interpretation of results.

Overall, the major findings are not particularly unexpected, though some of the details are thought provoking. I am mostly left wondering how these mutational processes influence the biology of the immune system, which limits the impact of the paper somewhat. I realize that that this may be out of the scope of this work, but the most "interesting" questions would stem from understanding the downstream consequences of the mutations. Recombination, somatic hypermutation, and class switching were all positively selected during vertebrate evolution. What are the tradeoffs for this mutational burden (besides cancer risk, which is already well known)? How does human genetic variation influence the accumulation of these mutations, and would there be genetic correlation between loci influencing mutation rate and autoimmune disease or immune response? The observation about the UV damage signature in lymphocytes is interesting, and potentially relevant clinically, as photopheresis is frequently used as a treatment for GVHD and PTCL. Same with the GI signature, but these are not followed up with additional analysis. The authors do a great job describing the phenomena, but it leaves me wanting more explanation about the impact on biology.

Referee #3 (Remarks to the Author):

This is an elegant study looking at mutations and structural variations in various lymphocytes (naive and memory B/T cells). The authors isolated specific lymphocytes and performed single-cell expansions of these cells to obtain sufficient cells and DNA for whole genome sequencing. 717 whole genomes were sequenced and analyzed. DNA from hematopoietic stem cells was used as control.

The authors find that all lymphocyte cells had more mutations and structural variations than the hematopoietic stem cells, and higher mutations in memory cells compared to naive cells. Off-target effects of immunological diversification account for most of the additional mutations. Analysis of the mutational signatures provided insight in the origin of the mutations (deamination of methylated cytosines, UV light damage, microenvironmental exposure, etc.). Analysis of the structural variations (non-IG/TCR rearrangements) revealed that memory cells had more SVs than naive cells, deletions were most frequent (some involving known tumor suppressor genes) and most were the result of off-target VDJ recombination events. By comparing these data from normal lymphocytes with data from malignant cells (leukemia/lymphoma), the authors conclude that transformation of lymphocytes to malignant cells occurs on the basis of the same mutational processes active in normal lymphocytes.

The data are presented in a clear and balanced way, and provide novel and relevant insight in the mutational processes that are active in normal lymphocyte development.

I only have a few minor remarks:

Figure 1B shows the mutation frequency for naive B, memory B, naive T and memory T cells. The color legend is somewhat confusing, as I expected that each plot showed the data for all the cell types, but there are 4 different plots showing the data for one of each cell types. I would suggest to use only 2 colors purple and green ; grey for the other cell types.

page 5: HSPCs accumulated 16 mutations/cell/year other cell types 15, 22, 25 mutations/cell/year: is it possible to correlate this with mutations per cell division ?

Comparison with blood cancer data: the sequence data from the normal lymphocytes are based on 20x whole genome sequence data, is that sufficiently deep to compare with cancer genome sequence data at >100x : can the authors discuss any limitations of the 20x sequence coverage?

Author Rebuttals to Initial Comments:

Referees' comments:

Referee #1 (Remarks to the Author):

The manuscript by Machado et al. examines the accumulation of mutations in human lymphocytes across their development. To accomplish this, the research team performed whole genome sequencing on colonies derived from sorted naive and memory B and T lymphocytes. The authors have previously optimized protocols for expanding flow-sorted single lymphocytes and have convincingly demonstrated that majority of mutations detected in these colonies accurately represent genomic events in the sorted cells. The accumulation of mutations is evaluated as a function of age of the donor and stage of lymphocyte differentiation (naive vs. memory). The findings, that mutations accumulate as a function of lymphocyte diversification process and that lymphocytes have significantly more mutations than HSCs is fundamental to our understanding of the immune cell development and lymphomagenesis. Few groups are able to do WGS of individual lymphocytes without genome amplification and the researchers perform in depth analysis on the valuable data in hand from their sequencing efforts as well as prior sequencing of lymphoma cells to examine the contribution of mutations and off-target rearrangements to lymphomagenesis. I believe the presented work is of interest to the broader audience and is appropriate for the journal.

Computational methods are well described, manuscript reads well and is easy to understand. Conclusions are drawn based on sequence analysis of >70 colonies per cell type with appropriate statistical tests utilized. There are a few analysis/comparisons that are well within grasp given the data in hand that would further strengthen the manuscript.

We thank the reviewer for these generous comments.

For B cells:

- A lot of analysis that infers past GC experience is done on sorted memory B cells. Can the authors (1) confirm that majority of these lymphocytes have undergone CSR - not just via IgD- gating but sequence analysis of the constant region (2) evaluate SHM load in the intronic region of Ig rearrangement to compare mutation load at the Ig loci vs. other chromosomes? Figure 3B shows the increase in SBS9 signature in the recombined IgH region but it unclear precisely how this burden compares to other genomic regions, and what the spatial distribution of these mutations is.

Supplementary figure S7 suggests that there is a significant increase on Ch14, presumably at IgH - but whether this is restricted to productive Ig rearrangements, is on both alleles and what the distribution is across Igk and Igl should be shown.

These are excellent suggestions to further characterise class-switch recombination (CSR), the distributions of mutation rates across Ig loci, and the effects of somatic hypermutation (SHM) on productive versus non-productive Ig rearrangements. We have addressed each of these points below:

- To determine the CSR status of B cell genomes, we combined output from the algorithm IgCaller¹ (designed to call V(D)J and CSR rearrangements from deep targeted sequencing of lymphomas), the frequency of split reads reporting the CSR recombination, and the read depth across class-switch regions for inferring copy number. Reassuringly, we do indeed find that a majority of the memory B cell genomes show evidence of class switch recombination (76%), compared to 12% of naive B cells having undergone CSR. As expected, the distribution of isotypes was predominantly IgG (66%; IgG1 41%, IgG2 4% and IgG3 21%) and IgA (34%; IgA1 23% and IgA2 11%). **We have now reported the frequency and distribution of B cell CSR events in the main manuscript ('Structural variants', p. 9); added a new supplementary table (Table S6) containing the CSR status and locus details for each B cell genome; and detailed our methods for determining CSR status in the Supplementary Methods.**
- We have analysed the genomic distribution of the SBS9 and SHM signatures in more detail. Interestingly, although the number of canonical SHM mutations (percent of the productive V gene mutated) and the overall amount of SBS9 in a given B-cell genome are correlated ($p=4 \times 10^{-9}$; **Figure 3B**), SBS9 is not particularly enriched at the Ig loci; only the canonical SHM signature is. The strongly significant correlation suggests that SBS9 occurs during the germinal centre reaction, but the lack of enrichment at Ig loci (and different mutational spectrum) suggests it is a different process to canonical SHM. **We have now more explicitly commented on the different genomic distributions of SBS9 and Canonical SHM in the main text ('Signatures of the germinal centre reaction', pp. 6). We have also added a new panel to Extended Figure 7 (panel D, pasted below as Reviewer Figure 1) showing the distribution of SBS9, SBSblood and canonical SHM signatures across chromosomes 2, 14, and 22, colouring the locations of the Ig loci.**
- We performed a detailed analysis of the patterns of mutation density across Ig rearrangements compared to non-rearranged Ig loci and non-Ig loci. This analysis revealed a number of interesting observations.
 - In memory B cells, the mutation burden of V genes from productive V(D)J rearrangements was much greater than that of V genes from non-recombined V genes (Figure S8A; mean of 7×10^{-3} and 3×10^{-4} variants/bp/genome for productive and non-recombined V gene exons respectively, t-test $p=1.2 \times 10^{-8}$). Even so, the mutation rate at non-recombined V genes was still significantly greater than the genomic background (mean 3×10^{-4} versus 2×10^{-6} for non-recombined V gene exons and non-Ig exons respectively, t-test $p=5.2 \times 10^{-7}$). **These data are now discussed in the manuscript ('Signatures of the germinal centre reaction', p. 6) and shown in a new Extended Figure 7 (panel A), pasted below as Reviewer Figure 2.**
 - We can attribute this increased mutation burden to the effects of canonical SHM rather than V(D)J recombination, as the mutation burden of productive V genes in naive B cells has a mean 3×10^{-4} variants/bp/genome, much lower than that of memory B cells, and similar to that of the

non-recombined V genes in memory B cells (now shown in Extended Figure 7A, pasted below as Reviewer Figure 2).

- This SHM-related increase in mutation burden was observed in both exons and introns, and across kappa, lambda and heavy chain regions of the genome. We have shown the distributions across the various Ig regions in the new Extended Figure 7B-C, pasted below as Reviewer Figure 3.

Reviewer Figure 1 (included as Extended Figure 7D). Proportion of mutations across chromosomes 2, 14 and 22 in each 1Mb window attributed to signatures SBS9, SBSblood and the canonical somatic hypermutation (SHM) signature (rows). Windows spanning the relevant immunoglobulin regions are coloured according to the key.

Reviewer Figure 2 (included as Extended Figure 7A). Estimates of the mutation rate across non-Ig chromosomes and Ig regions for memory (left) and naive (right) B cells. Rates for the Ig regions are calculated separately for the productive (triangles) and non-recombined alleles (circles) and exons (green) versus introns (orange).

Reviewer Figure 3 (included as Extended Figure 7B-C). Estimated mutation rates across different variable segments of the Ig genes for exons (green) versus introns (orange) for productive (top row) and non-recombined alleles (middle row). Number of productive V(D)J rearrangements affecting each variable segment in the dataset (bottom row).

- It is not immediately clear why the authors expect 1-2 V(D)J events per lymphocyte, as each T or B cell would have experienced at least 2 (IgH and IgL; TCRa and TCRb) and possibly more if the initial recombination event is not successful. So this sentence seems a bit out of tune: "We found 1037 SVs across 635 lymphocytes, of which 85% occurred in Ig/TCR regions, consistent with 1-2 V(D)J recombination events per lymphocyte"?

The reviewer is absolutely correct in this reasoning – we would expect at least two V(D)J events per lymphocyte genome, and believe that we are under-calling a fraction of structural variants in our dataset. There are several factors contributing to this, including the highly repetitive nature of these loci, the frequency of germline polymorphism, the density of canonical SHM mutations near structural variants impairing read mapping and the variable sequencing depth we achieved (linear regression of number of missing V(D)J recombination events versus coverage: $R^2=0.17$, $p=2 \times 10^{-16}$). We estimate our overall sensitivity to detect V(D)J events is ~62%. For the reasons discussed above, V(D)J

rearrangement is considerably more difficult to detect than structural variants in other regions, so this estimate is likely a lower bound for the genome-wide sensitivity for SVs. Unfortunately, due to the relatively small number of SVs, we cannot perform the same sort of mutation burden depth correction that we run for the SNVs. We are confident that the key points about SV burden are not affected by this reduced sensitivity. **We have edited the manuscript to clarify this point, and to acknowledge the reduced sensitivity for SVs in the immunoglobulin loci ('Structural variants', p. 8-9).**

For analysis of T cells:

- The authors demonstrate the B cell malignancies harbored a comparable number of mutations to "normal" B lymphocytes with a dominant SPS9 signature, but had elevated number of structural variation. The authors should examine T cell lymphomas (and in particular cutaneous T cell lymphomas) in light of the UV signature found in some "normal" memory T cells. WGS has previously be performed in CTCL and similar analysis can be done in data from CTCL vs. memory T cells - to examine the nature and frequency of mutations and SVs.

This is an excellent suggestion, and we have now incorporated 5 published cutaneous T-cell lymphoma (CTCL) WGS samples² into our analysis. Interestingly, as implied by the reviewer, we do find that a high proportion (20-70%) of base substitutions in these samples is accounted for by the ultraviolet signature (SBS7a), further supporting the notion that skin-resident T cells do experience the mutational pressures associated with sun exposure. As seen for the B-cell malignancies, the number of structural variants in the CTCL genomes far exceeded those observed in the normal T cells. **We include CTCL in the plots comparing mutation burden and SV burden (Fig. 5A-B), mutational signatures (Fig. 5C), and RAG deletion rate (Fig. 5H). We have also performed a new analysis comparing SBS7a proportion across cell types and malignancies (Extended Fig. 6C, pasted below as Reviewer Figure 4).**

Reviewer Figure 4 (included as Extended Figure 6C). Proportion of mutations attributable to SBS7a across normal lymphocytes and lymphoid malignancies. Boxes show the interquartile range and the centre horizontal lines show the median. Whiskers extend to the minimum of either the range or 1.5× the interquartile range.

For both T and B:

- Total mutation "burden" would be easier to appreciate if it is plotted across genome for each cell type with Ig and TCR alleles highlighted. This may make it easier for the reader to appreciate existence of any specific hotspots and relative frequency compared to mutations associated with VDJ rearrangement and SHM. Similar to what is done in figure S7 for the SBS9 signature
- Same for structural variation

This is an excellent suggestion – we have now plotted SNV and SV burdens across the genome, highlighting the Ig and TCR alleles, now included as Extended Figure 10A-B, pasted below as Reviewer Figure 5.

Reviewer Figure 5 (included as Extended Figure 10A-B). (A) Mutation rates across the genome for SNVs split by different cell types, with chromosomes labelled in the top strip, and Ig/TCR regions marked. (B) Structural variation rates across the genome split by different cell types, with chromosomes labelled in the top strip, and Ig/TCR regions marked.

General minor points:

- In discussing data in Fig S2 - please state the indel rate in B and T cells when comparing to HSC. Currently only the indel rate in HSCs is given.

We have added the indel rates for lymphocytes to the main text ('Mutation burden', p. 4).

- The statement pertaining to association of telomere length on pg. 8 appears to be a bit bold "These data confirm that telomeres do lengthen in the germinal centre reaction, and provide further evidence that off-target SBS9 mutations are generated in the germinal centre." as the analysis was done on memory B cells, and while many may have gone through the GC reaction the authors have not demonstrated that.

We have softened this statement to reflect the uncertainty ('Signatures of the germinal centre reaction', p. 6).

Lastly, presumably 20x coverage for somatic mutation was, at least in part, based on practical/financial considerations. The analysis is "benchmarked" against HSCs. However, given the long-standing convention that 30x coverage minimum is required for confident mutation calling, the authors should perhaps address that in a sentence within the text...

This paper compares SNV calling from 2x-100x and found 30x was the minimum for high confidence:

<https://www.ncbi.nlm.nih.gov/pmc/articles/PMC5591230/> [ncbi.nlm.nih.gov]

Illumina recommendations:

https://www.illumina.com/Documents/products/technotes/technote_snp_caller_sequencing.pdf [illumina.com]

This is an interesting point about the depth of coverage and its effect on mutation detection.

Our experimental design of sequencing many single-cell-derived colonies from each subject enables us to accurately estimate our sensitivity for calling single nucleotide substitutions – essentially, because the colonies are single-cell-derived and lymphocytes are diploid, somatic mutations are present at 50% variant allele fraction, the same as heterozygous germline SNPs. Having sequenced tens to hundreds of colonies from the same individual, we can straightforwardly identify a highly accurate set of true heterozygous germline SNPs for that subject. We can then assess what proportion of this 'truth set' of germline SNPs is called by our algorithm in each colony, versus sequencing depth in that colony.

Two reassuring observations emerge from this analysis (see **Reviewer Figure 6** below):

- We find that our sensitivity for detecting single nucleotide substitutions follows a logistic curve, with high sensitivity in the range of coverage achieved for the colonies sequenced here. At 10x coverage, sensitivity for SNVs is ~80%; and by 20x coverage, more than 98%.
- There is negligible spread of estimated sensitivity levels for individual colonies around the mean (logistic) curve – in other words, the sensitivity for a specific colony can be predicted extremely accurately just from its mean sequencing depth. This means that there are no

additional factors such as GC bias, unevenness of coverage or batch effects that influence our sensitivity, and we can therefore correct mutation burdens for the estimated sensitivity with confidence.

We used this information to inform our depth-based mutation burden correction, which corrects the somatic substitution and indel burdens per genome using an asymptotic regression of mutation burden with coverage across HSPC genomes of a given donor. **We have now discussed these analyses of sensitivity in the main text ('Genome sequencing of B and T lymphocytes', p. 3), included the figure showing the relationship between coverage and sensitivity as Extended Figure 3B (pasted below as Reviewer Figure 6) and updated the Methods with the details of this analysis.**

An important additional point, also noted by reviewer 3, is that the PCAWG blood cancer samples were sequenced at higher coverage than our colonies. However, both normal cell contamination and increased ploidy of the tumour cells mean that the effective coverage of the cancer samples had an overall comparable distribution to our normal lymphocyte colonies (shown in **Reviewer Figure 11**, in the response to reviewer 3). Therefore, the comparison of mutation burden between normal lymphocytes and blood cancers reported is based on similar values for relevant sequencing coverage. **These data are reported in the main text ('Comparison with malignancy', p. 9) and the histogram comparing normal and cancer depths is included as Extended Figure 10C.**

Reviewer Figure 6 (included as Extended Figure 3B). Estimates of sensitivity for mutation calling as a function of depth for each colony (points in left panels) from each donor (rows). The second column of panels shows uncorrected estimates of mutation burden for HSCs in each donor, while the third column shows mutation burden estimates after correction for sequencing depth by asymptotic regression. The fourth column shows the corrected mutation burdens for lymphocyte colonies.

Referee #2 (Remarks to the Author):

In this study, Machado et al. set out to comprehensively determine the mutations rate in human T- and B- lymphocytes at various stages of development and in different tissues. They obtained tissue from 5 sources (marrow, cord blood, peripheral blood, spleen, and tonsil) and 8 subsets were sorted, and 6 cell types were grown into colonies used for WGS (naïve and memory T and B-cells, T reg, and

HSPCs). Novel culturing methods were developed to allow for in vitro expansion. They find that mutation burden associates with increasing age and cell type, with memory cell types having a higher burden than naïve. Mutational processes were similar in naïve lymphocytes and HSPCs, but was greater variability in mutational burden in memory cells. This was attributable largely to the mutations introduced in the germinal center reaction in the IGHV locus (SBS9), but also additional novel signatures contributed (UV light and GI exposure). SBS9 was associated with late replicating, closed chromatin regions of the genome, so may be related to error-prone bypass by polymerase η . Indels in lymphocytes were associated with RAG activity, but not class switching. They further find that increased mutational burden in some lymphoid malignancies is due to heightened mutation rate, but with the same processes as in normal lymphocytes.

The paper is well written and technically a tour-de-force on mutational processes in lymphocytes. The conclusions are mostly justified. Since the paper is entirely descriptive, some of the conclusions would be strengthened by experimental validation, for example the polymerase η bypass hypothesis for SBS9. The development of culturing methods for human lymphocytes seems to be a new advance, but requires much more detail so others can replicate. There should also be a supplemental method describing the results of the expansion for each cell type from each donor. How efficient was the expansion? Did a large proportion of cells not form colonies? If so, this could bias the results of the sequencing and influence the interpretation of results.

We thank the reviewer for these favourable comments and for emphasising the importance of the novel protocols for producing single-cell-derived B and T cell colonies. Clearly, our results are only generalisable if the lymphocyte colonies we have sequenced are representative of the wider pool of lymphocytes within the individual research subjects. To address this question, we have undertaken three reanalyses of our data, all now reported in the manuscript: (1) estimating the efficiency of the culture system for each cell type; (2) comparing clonality of mutations in cultured lymphocytes to bulk populations from the donors; and (3) assessing the flow cytometric measurements of lymphocytes that successfully seeded colonies and those that did not.

Efficiency of culture system

The culture efficiencies we achieved ranged from 0.5% to 14%, depending on the cell type. The naïve CD4⁺ T cells had the highest efficiency at 14%, while the lowest efficiency cell type was memory CD8⁺ T cells, with an average efficiency of 0.5%. As a result, these latter cells are rare in our analysis (a total of 8 cells out of 635 lymphocytes) and represent 8% of the memory T cells sequenced, the remaining 92 being CD4⁺ memory T cells. The remaining cell types had efficiencies in the range of 2-5% (naïve B, 2%; memory B, 2%; memory CD4⁺ T, 5%; naïve CD8⁺ T, 5%; Tregs, 3%).

We have added a detailed protocol for these culture procedures in the Supplementary Methods section that provides a step-by-step process for generating the colony expansions to enable our

study to be reproduced in other laboratories. Here we also list the net culture efficiency (proportion of sorted cells that grew to ≥ 30 cells and were picked for sequencing), and we have added a Supplementary Table S1 containing the culture efficiencies for each of the different FACS experiments.

Clonality of cultured lymphocytes versus bulk populations

Efficiency rates of 2-5% in our culture system mean that we are accessing a sizable subset of the whole lymphocyte population within an individual, given the large total population size of circulating lymphocytes. Nonetheless, there remains the possibility, as the reviewer raises, that the population that does successfully grow in culture is biased in some way – perhaps they preferentially derive from fitter clones or from cells with particular sub-lineage biases, for example.

To address whether there might be clone-to-clone biases in which cells successfully seeded colonies, we studied the phylogenetic relationships among the cells from a given research subject. First, none of the lymphocytes sequenced shared a CDR3 sequence, indicating that we did not sequence multiple colonies derived from the same *in vivo* clonal expansion. Second, we leveraged the fact that one individual in our current cohort (AX001) was the same person as that enrolled in our earlier study of HSC dynamics, where we performed deep targeted resequencing of bulk B and T cell populations³. For this individual, we compared the fraction of lymphocyte colonies reporting a given somatic mutation with the variant allele fraction of that mutation in the bulk resequencing data. Reassuringly, we find a very high correlation between these variant allele frequencies (B cells, $R^2=0.93$, $p=2 \times 10^{-16}$; T cells, $R^2=0.96$, $p=2 \times 10^{-16}$; linear regression; see **Reviewer Figure 7** below). This suggests that there is negligible bias arising from variation among clones in propensity to seed *in vitro* colonies.

We have included these clonality assessments in Supplementary Methods, and signposted them in the main text ('Genome sequencing of B and T lymphocytes', p. 3). The panels below from Reviewer Figure 7 have been included as Extended Figure 3A.

Reviewer figure 7 (included as Extended Figure 3A). Fraction of colonies (x axis) in AX001 reporting somatic mutations versus the variant allele fraction of that mutation (y axis) in B cells (left panel) and T cells (right panel).

Flow cytometric parameters of successful versus unsuccessful colonies

The other approach we undertook to evaluate potential sources of bias in colony growth was to leverage the index-sorting data of our single cells. This enabled us to compare the cell surface expression of various markers between cells seeding colonies picked for sequencing and sorted cells that did not grow colonies. Reassuringly, for naive and memory B cells, there were no significant differences between successful and unsuccessful cells in fluorescence intensity of CD19, CD20, CD27 or IgD for any of the flow-sorts in any of the research subjects (see **Reviewer Figure 8** below). We did find some significant differences in fluorescence intensity for T cells between those that successfully grew and those that did not. However, these were not consistent across different individuals, nor were they always in the same direction. One difference, seen in two research subjects, was that T cells higher in CCR7 were somewhat more likely to generate successful colonies in culture. For naive T cells, we gated for CCR7^{high}, attempting to exclude terminally differentiated effector T cells that are CD45RA^{high} (like the naive T cells) but CCR7^{low}. It is possible, however, that the CCR7^{high} gate for naive T cells still captured some terminally differentiated effector cells, which may not proliferate in culture. For the memory T cells, we included the entire CCR7 gate, attempting to culture both effector and central memory T cells. The CCR7^{high} central memory T cells grew somewhat better than the CCR7^{low} effector memory cells. Reassuringly, though, we nonetheless successfully cultured representative numbers of CCR7^{low} colonies, as can be seen in the overlaid data points on the Figure below.

Taken together, then, the index flow-sorting data suggest that any biases in colony efficiency within sorted populations are negligible.

We have referenced the comparison of index-sorting fluorescence intensity data between cells that successfully seeded colonies and those that did not in the main text ('Genome sequencing of B and T lymphocytes', p. 3); included Reviewer Figure 8 below as Extended Figure 2; and described the analysis in Supplementary Methods.

Reviewer figure 8 (included as Extended Figure 2). (A) Comparison of fluorescence intensity for colonies that successfully seeded colonies (red) versus those that did not (grey) for a representative flow-sorting experiment. (B) Box and whisker plots showing the range of fluorescence intensity levels for cells that successfully grew colonies (teal-green, with individual data points from each cell overlaid) versus those that did not (salmon-pink), broken down by research subject and date of flow-sorting (rows) and cell type (columns).

Overall, the major findings are not particularly unexpected, though some of the details are thought provoking. I am mostly left wondering how these mutational processes influence the biology of the immune system, which limits the impact of the paper somewhat. I realize that that this may be out of the scope of this work, but the most “interesting” questions would stem from understanding the downstream consequences of the mutations.

Recombination, somatic hypermutation, and class switching were all positively selected during vertebrate evolution. What are the tradeoffs for this mutational burden (besides cancer risk, which is already well known)?

This is an intriguing question. Somatic mutations arising from off-target physiological mutational processes could have deleterious or advantageous effects on lymphocytes, reducing or increasing the clone’s fitness respectively. To assess this, we measured the dN/dS ratio, a parameter to infer patterns of selection widely used in evolutionary genetics^{4,5}, recently adapted for cancer and somatic mutations⁶. Essentially, on the basis that synonymous mutations evolve neutrally, we can estimate the number of non-synonymous mutations expected by chance from this selectively neutral background mutational process. A dN/dS ratio of 1 implies either absent or exactly balanced positive and negative selection; a dN/dS ratio less than 1 implies a balance tilted towards negative (purifying) selection and a dN/dS ratio greater than 1 implies an excess of positive selection. In the cancer setting, this approach has typically been applied on a gene-by-gene basis to identify novel cancer genes⁷, but when applied to the whole exome, it enables a more global assessment of selection patterns⁶.

We estimated the exome-wide dN/dS ratio across all lymphocytes, excluding the immunoglobulin regions, to be 1.12 (CI_{95%}=1.06-1.19). This implies that positive selection exerts a considerably greater influence on lymphocytes than purifying selection, with approximately 11% (CI_{95%}=6-15%) of non-synonymous mutations conferring a selective advantage. The dN/dS estimates were broadly comparable across lymphocyte subtypes (albeit with wider confidence intervals – see figure below). Furthermore, the ratio remained elevated even when we excluded known cancer genes (1.12, CI_{95%}=1.06-1.18), suggesting that this positive selection acts on a broader set of genes than currently identified.

Although we do not have a comparable measure for selection acting on structural variation, we did observe several rearrangements hitting known tumour suppressor genes for lymphoma (such as *CREBBP*, *ATF7IP* and *ILF3*; see **Figure 4B** and **Table S5**). This could suggest a role for positive selection acting on structural variants in normal lymphocytes in addition to that seen for point mutations.

Overall, then, our data reveal that positive selection acting on somatic mutations in lymphocytes is a more pervasive effect than negative selection. This suggests that clonal expansions of individual lymphocytes would be the most important evolutionary trade-off for physiological genome editing during lymphocyte differentiation. As the reviewer indicates, lymphoid cancers are clearly one such consequence – the fact that mutation burdens and signatures of normal lymphocytes are similar to those seen in several lymphoid malignancies argues that this off-target mutagenesis is sufficient for occasional lymphocytes to transform.

Cancer may not be the only consequence of pervasive positive selection. For 50+ years, scientists have noted the similarity between the self-perpetuating lymphoid reaction of autoimmunity and that of lymphoid malignancy^{8,9}, leading to the hypothesis that driver mutations could underpin the capacity of autoreactive lymphocytes to evade normal suppression mechanisms¹⁰. Indeed, recent data in Sjögren's syndrome found driver mutations in known lymphoma genes in lymphocytes responsible for mixed cryoglobulinaemic vasculitis from four patients¹¹. Although we have studied lymphocytes from healthy individuals here, our data would certainly support such a mechanism for autoimmune disease – not only are mutation rates high enough to generate sufficient genetic diversity within lymphocytes, it appears that the local microenvironment is capable of invoking selective pressures that favour lymphocytes with particular drivers.

We have included the results from the exome-wide dN/dS analysis across the lymphocyte subsets in the main text ('Mutation burden', p. 4) and an extra panel as Extended Figure 4B. We have included more discussion of the potential downstream trade-offs in the Discussion (pp. 10-11), including possible links to malignancy and autoimmunity.

How does human genetic variation influence the accumulation of these mutations, and would there be genetic correlation between loci influencing mutation rate and autoimmune disease or immune response?

The hierarchical experimental design we have adopted in this and other studies, with multiple samples per donor, enables us to estimate both the within-person and between-person variances in mutation burden. The linear mixed effects model applied to our lymphocyte dataset estimates the cell-to-cell standard deviation of mutation burden *within* a donor to be 820 SNVs/cell for memory B and 592

SNVs/cell for memory T lymphocytes. By contrast, the estimate for person-to-person standard deviation in mean mutation burden was only 60 SNVs/cell. Thus, the within-person standard deviation is ~10-fold higher than the between-person variation.

This has two interesting corollaries. First, in the nature-vs-nurture dichotomy, it suggests that lifelong environmental forces (infections, vaccinations, inflammation, skin residency: 'nurture') are likely to exert a considerably stronger influence on the variability of mutation rates in lymphocytes than inherited genetic variation ('nature'). Second, considering potential downstream consequences of somatic mutations, these data suggest that the risk of an individual lymphocyte acquiring the driver mutations needed to promote a rogue clonal expansion will predominantly be shaped by the specific life history and exposures of that lymphocyte rather than a donor-wide set-point for mutation rates.

This, of course, does not argue that inherited genetic variation plays no role in influencing mutation rates in lymphocytes during differentiation. However, the sample sizes that would be required in a QTL GWAS to identify inherited SNPs influencing mutation rates would be massive given the considerable acquired variability of these rates.

We have included more details on the estimates for cell-to-cell and person-to-person variation in mutation rates in the Results ('Mutation burden', p. 4), and discussed these implications further in the Discussion (p. 10).

The observation about the UV damage signature in lymphocytes is interesting, and potentially relevant clinically, as photopheresis is frequently used as a treatment for GVHD and PTCL. Same with the GI signature, but these are not followed up with additional analysis. The authors do a great job describing the phenomena, but it leaves me wanting more explanation about the impact on biology.

We agree that the UV signature in lymphocytes is interesting and unexpected. To assess its impact on biology, we have downloaded and reanalysed published whole genome sequences from 5 cutaneous T cell lymphomas² (CTCLs). The overall mutation burden in the lymphomas ranged from a few hundred to a few thousand SNVs, with all samples showing the UV light signature, SBS7a, accounting for 20-60% of mutations. Interestingly, although mutation burdens varied extensively across the CTCLs, the overall numbers of mutations attributable to UV light were broadly comparable between the lymphomas and the normal memory T lymphocytes with high burden of SBS7a. As seen for the B-cell lymphomas, the numbers of structural variants in the CTCLs were considerably higher than the normal T lymphocytes.

Taken together, these data underscore one of the key themes of the manuscript. Although the mutational process here is exogenous, in the form of UV light, we find that normal T lymphocytes can acquire similar numbers of point mutations from this exposure as occur in malignant cutaneous T cells – a similar observation to that made for the rates of off-target physiological mutation processes in memory B cells. This emphasises that the processes generating point mutations in normal lymphocytes can generate sufficient somatic variants for progression towards malignancy, although full transformation often requires additional large-scale genome rearrangement.

We have included the CTCL data in the Results ('Comparison with malignancy', p. 9). We include CTCL in the plots comparing point mutation burden and SV burden (Fig. 5A-B), mutational signatures (Fig. 5C), and RAG deletion rate (Fig. 5H). We have included a panel showing the comparison of SBS7a proportion across cell types and malignancies (Extended Fig. 6C, included as Reviewer Figure 4 above).

Referee #3 (Remarks to the Author):

This is an elegant study looking at mutations and structural variations in various lymphocytes (naive and memory B/T cells). The authors isolated specific lymphocytes and performed single-cell expansions of these cells to obtain sufficient cells and DNA for whole genome sequencing. 717 whole genomes were sequenced and analyzed. DNA from hematopoietic stem cells was used as control.

The authors find that all lymphocyte cells had more mutations and structural variations than the hematopoietic stem cells, and higher mutations in memory cells compared to naive cells.

Off-target effects of immunological diversification account for most of the additional mutations.

Analysis of the mutational signatures provided insight in the origin of the mutations (deamination of methylated cytosines, UV light damage, microenvironmental exposure, etc.).

Analysis of the structural variations (non-IG/TCR rearrangements) revealed that memory cells had more SVs than naive cells, deletions were most frequent (some involving known tumor suppressor genes) and most were the result of off-target VDJ recombination events.

By comparing these data from normal lymphocytes with data from malignant cells (leukemia/lymphoma), the authors conclude that transformation of lymphocytes to malignant cells occurs on the basis of the same mutational processes active in normal lymphocytes.

The data are presented in a clear and balanced way, and provide novel and relevant insight in the mutational processes that are active in normal lymphocyte development.

I only have a few minor remarks:

Figure 1B shows the mutation frequency for naive B, memory B, naive T and memory T cells. The color legend is somewhat confusing, as I expected that each plot showed the data for all the cell types, but there are 4 different plots showing the data for one of each cell types. I would suggest to use only 2 colors purple and green ; grey for the other cell types.

We have edited this figure, as suggested, to a 2-colour scheme (purple for lymphocytes and green for HSPCs; grey for other cell types) for improved clarity (as well as the equivalent figure for indels, Extended Figure 4A).

page 5: HSPCs accumulated 16 mutations/cell/year other cell types 15, 22, 25 mutations/cell/year: is it possible to correlate this with mutations per cell division?

This is certainly an interesting question – to what extent are the differences in mutation rates across cell types explained by differences in cell division rates or differences in DNA damage and repair processes? We cannot provide a definitive answer, but we can offer some insights from further analysis of our data.

Clock-like activity of signature SBS1 in lymphocytes

While it is true that many mutational processes are tied to DNA replication (such as DNA polymerase errors), some processes must be active in interphase. The observation that even post-mitotic cells such as neurones continue to accumulate somatic mutations at a linear rate throughout life^{12,13} argues that not all mutational processes are coupled to cell division. We estimated the rates of the clock-like mutational processes separately for each signature across our dataset. Three mutational signatures correlated with age across cell types – SBS1, SBSblood and SBS8. Interestingly, all three signatures exhibited significantly higher rates per cell per year in T cells than in naïve B cells and HSPCs (see **Reviewer Figure 9** below).

Of particular relevance to whether differences across cell types are linked to different cell division rates is the SBS1 signature. This signature, characterised by C>T mutations at CpG dinucleotides, is caused by deamination of methylated cytosine directly to thymine. It is a spontaneous hydrolytic

reaction in dsDNA occurring throughout the cell cycle¹⁴, with the mismatch repair pathway repairing >90% of such events. Since the deamination generates a thymine directly, the G:dT mismatch must be repaired *before* replication: otherwise the dT becomes fixed as a mutation once replicated. Interestingly, in cells from individuals constitutively deficient in mismatch repair, there is no increase in SBS1 after neoplastic transformation¹⁵, consistent with the rate of DNA damage (deamination) being largely independent of replication. Moreover, the tumour types with the highest rates of SBS1 in sporadic mismatch repair deficiency are those from tissues with low intrinsic cell division rates, such as meningiomas, mesotheliomas and gliomas¹⁶. This argues that the tug-of-war between cytosine deamination and repair leading to SBS1 mutations predominantly plays out during interphase rather than being coupled to replication.

The observation that SBS1 rates were higher in T cells therefore argues that differences in cell division rates across cell types do not fully explain differences in mutation rates. **We have included the analysis of mutation rates for different signatures across cell types in the Results ('Mutational signatures', p. 5), and the figure below as Extended Figure 5.**

Reviewer figure 9 (included as Extended Figure 5). Mutation rates with age for the different mutational signatures identified. (A) Scatterplots showing estimated numbers of mutations attributable to each signature (y axes) versus age in years (x axis), coloured by cell type. Lines show estimated slopes for each cell type, with shaded areas representing 95% confidence intervals for the

slope. (B) Estimates of the rate per year of SBS1 (green), SBSblood (blue) and SBS8 (pink) across cell types. Circles denote point estimates, with error bars 95% confidence intervals.

Telomere attrition across cell types

A second approach to inferring cell division rates is to study telomere attrition, since this is explicitly linked to the end-replication problem of linear chromosomes – it is thought that telomeres only shorten with cell division. *In vitro* studies of telomere attrition in lymphocytes have estimated that telomeres shorten by ~50-100bp per cell doubling on average, at similar rates for naïve and memory T cells¹⁷⁻¹⁹. This is broadly similar to the 30-100bp loss per cell doubling estimated for HSCs²⁰.

We have estimated telomere lengths for each colony in our dataset from the whole genome sequencing data. We find that for HSPCs, naïve B cells, naïve T cells and memory T cells, telomere lengths decreased steadily with age, as expected. The rates of telomere attrition were broadly comparable across these cell types ($p=0.09$ for differences in slope across cell types), at ~44bp per year (see **Reviewer Figure 10** below; memory B cells were excluded from this analysis due to their increasing telomere length during the germinal centre reaction). These rates estimated from whole genome sequencing data are remarkably consistent with previously published estimates of telomere loss with age of 30-50bp per year using radioassays for terminal restriction fragment lengths^{17,19}.

Taken together, these telomere data suggest that T lymphocytes divide about once a year in the adult maintenance phase. Direct assays using isotope labelling have provided similar estimates of a cell division every 3-24 months, with memory T cells more rapid than naïve T cells²¹. This implies that mutation rates are approximately in the range 5-50 mutations per cell division for T lymphocytes.

We have included the figure below as Extended Figure 8 showing the rates of telomere attrition with age across cell types. The implications are discussed in the Results ('Signatures of the germinal centre reaction', p. 6) and Discussion (p. 10).

Reviewer figure 10 (included as Extended Figure 8). Scatterplots showing the estimated telomere lengths per colony (in base pairs; y axes) by age (in years; x axes).

Comparison with blood cancer data: the sequence data from the normal lymphocytes are based on 20x whole genome sequence data, is that sufficiently deep to compare with cancer genome sequence data at >100x : can the authors discuss any limitations of the 20x sequence coverage?

Since our colonies are single-cell-derived and come from diploid cells, a mean depth of 20x means that there will be an average of 10 reads reporting each variant. Importantly, the variant allele fractions will be equivalent for somatic mutations and heterozygous germline polymorphisms because the colonies are clonally pure. With multiple colonies sequenced per donor, we have a highly accurate catalogue of heterozygous germline SNPs – by measuring our recall rates for these ‘known’ SNPs for each colony, we obtain estimates of our sensitivity for calling somatic mutations. Studying this across colonies, we find that sensitivity for SNVs is ~80% at 10x coverage and >98% at 20x coverage, with the depth-versus-sensitivity curve reassuringly similar across donors and cell types (see Reviewer Figure

6 in the response to a related question from reviewer 1). These estimates for sensitivity suggest that our dataset is robust for estimating and comparing mutation burdens across cell types and donors.

As the reviewer indicates, the cancer genomes have deeper sequencing coverage than the normal lymphocytes. However, two of the reasons that cancer genomes need greater sequence depth is that they have (1) variable fractions of normal cell contamination that dilute the number of reads covering a variant; and (2) variable, often increased, ploidy levels that reduce the mean coverage per chromosome copy. Recent analyses of the PCAWG dataset have demonstrated that a strong predictor of sensitivity for calling mutations in cancer genomes is the mean number of reads per tumour chromosome copy²² – this metric can be calculated from the depth, ploidy and cellularity estimates for each cancer, and can be directly compared with the equivalent calculation for our normal lymphocytes (which have ploidy 2 and cellularity of 100%). This comparison shows that the effective coverage of the PCAWG blood cancer samples broadly overlaps with the effective coverage of the normal lymphocytes (see **Reviewer Figure 11** below). This suggests that the comparison of mutation burdens and signatures we report occurs on a relatively level footing.

We have now reported these estimates of sensitivity in the main text ('Genome sequencing of B and T lymphocytes', p. 3). We have included Reviewer Figures 6 and 11 showing coverage-sensitivity relationships for our data and the PCAWG genomes as Extended Figures 3B and 10C respectively. Finally, we have updated the Methods (p. 21) with the details of this analysis.

Reviewer Figure 11 (included as Extended Figure 10C). Histogram showing the distribution of effective sequencing depth per relevant chromosome copy from the PCAWG blood cancer genomes

(pink) and normal lymphocytes (blue). The number of reads per chromosome copy for both was calculated using the standard formula²²:

$$\text{nrpcc} = \frac{\rho\mu}{\rho\psi_T + (1 - \rho)\psi_N},$$

where μ is mean coverage; ρ is purity; and ψ_T and ψ_N are the ploidy of tumour cells and normal cells respectively. For cancer genomes, purity and ploidy were estimated from the copy number patterns; for lymphocyte colonies, the purity was 1 and ploidy was 2.

References

1. Nadeu, F. *et al.* IgCaller for reconstructing immunoglobulin gene rearrangements and oncogenic translocations from whole-genome sequencing in lymphoid neoplasms. *Nat. Commun.* **11**, 1–11 (2020).
2. McGirt, L. Y. *et al.* Whole-genome sequencing reveals oncogenic mutations in mycosis fungoides. *Blood* **126**, 508–519 (2015).
3. Lee-Six, H. *et al.* Population dynamics of normal human blood inferred from somatic mutations. *Nature* **561**, 473–478 (2018).
4. Nei, M. & Gojobori, T. Simple methods for estimating the numbers of synonymous and nonsynonymous nucleotide substitutions. *Mol. Biol. Evol.* **3**, 418–426 (1986).
5. Greenman, C., Wooster, R., Futreal, P. A., Stratton, M. R. & Easton, D. F. Statistical analysis of pathogenicity of somatic mutations in cancer. *Genetics* **173**, 2187–2198 (2006).
6. Martincorena, I. *et al.* Universal Patterns of Selection in Cancer and Somatic Tissues. *Cell* **171**, 1029–1041 (2017).
7. Rheinbay, E. *et al.* Analyses of non-coding somatic drivers in 2,658 cancer whole genomes. *Nature* **578**, 102–111 (2020).
8. Dameshek, W. & Schwartz, R. S. Leukemia and auto-immunization- some possible relationships. *Blood* **14**, 1151–1158 (1959).
9. Burnet, F. M. A reassessment of the forbidden clone hypothesis of autoimmune disease. *Aust. J. Exp. Biol. Med. Sci.* **50**, 1–9 (1972).
10. Goodnow, C. C. Multistep Pathogenesis of Autoimmune Disease. *Cell* **130**, 25–35 (2007).
11. Singh, M. *et al.* Lymphoma Driver Mutations in the Pathogenic Evolution of an Iconic Human Autoantibody. *Cell* **180**, 878–894.e19 (2020).
12. Lodato, M. A. *et al.* Aging and neurodegeneration are associated with increased mutations in single human neurons. *Science (80-.).* **559**, 1–8 (2017).

13. Abascal, F. *et al.* Somatic mutation landscapes at single-molecule resolution. *Nature* **593**, 405–410 (2021).
14. Shen, J. cheng, Rideout, W. M. & Jones, P. A. The rate of hydrolytic deamination of 5-methylcytosine in double-stranded DNA. *Nucleic Acids Res.* **22**, 972–976 (1994).
15. Sanders, M. A. *et al.* Life without mismatch repair. *bioRxiv* 2021.04.14.437578 (2021). doi:10.1101/2021.04.14.437578
16. Fang, H. *et al.* Deficiency of replication-independent DNA mismatch repair drives a 5-methylcytosine deamination mutational signature in cancer. *Sci. Adv.* **7**, eabg4398 (2021).
17. Vaziri, H. *et al.* Loss of telomeric DNA during aging of normal and trisomy 21 human lymphocytes. *Am. J. Hum. Genet.* **52**, 661–7 (1993).
18. Weng, N. P., Hathcock, K. S. & Hodes, R. J. Regulation of telomere length and telomerase in T and B cells: a mechanism for maintaining replicative potential. *Immunity* **9**, 151–7 (1998).
19. Weng, N. P., Levine, B. L., June, C. H. & Hodes, R. J. Human naive and memory T lymphocytes differ in telomeric length and replicative potential. *Proc. Natl. Acad. Sci. U. S. A.* **92**, 11091–11094 (1995).
20. Vaziri, H. *et al.* Evidence for a mitotic clock in human hematopoietic stem cells : Loss of telomeric DNA with age. *Proc. Natl. Acad. Sci. U. S. A.* **91**, 9857–9860 (1994).
21. Macallan, D. C., Busch, R. & Asquith, B. Current estimates of T cell kinetics in humans. *Curr. Opin. Syst. Biol.* **18**, 77–86 (2019).
22. Dentre, S. C. *et al.* Characterizing genetic intra-tumor heterogeneity across 2,658 human cancer genomes. *Cell* **184**, 2239–2254.e39 (2021).

Reviewer Reports on the First Revision:

Referees' comments:

Referee #1 (Remarks to the Author):

The authors have done an outstanding job of responding to reviewer concerns and have further improved this elegant manuscript. The newly added analysis and characterization provides further insight into biology of B and T cells and will be of great interest to immunologists, while the added experimental detail will certainly also be appreciated.

In all, I find that the authors provided one of the most comprehensive responses to reviewers that I have seen and this manuscript will be a benchmark for studies into mutations and structural variations in hematopoietic system.

Referee #2 (Remarks to the Author):

The authors now provide detail on the culturing methods and efficiencies, which shows that 0.5-14% are able to expand. This is considerably lower than their experiments on HSPCs, and could be a major source of confounding. To determine whether this leads to biased representation of cells in the sequence data, they perform three analyses.

First, they assess CDR3 regions of the sequenced cells. They do not find any redundant CDR3 sequences, suggesting that they are likely not expanding specific clones in vitro based on a TCR/BCR sequence. They also look at flow cytometric profiles of the sorted cells, which also confirms no evidence of bias by this metric. The final, and most important analysis, looks at the prevalence of specific mutations in colony expanded cells, versus in bulk cells from blood from one donor. This is the most important because it shows definitively whether the colony expanded cells are truly a random selection of all circulating cells based on proportional representation of clones in colonies vs blood. They plot the VAF of mutations in blood versus percent of colonies with the mutation in Ext Fig 3A. I believe there is an error in the figure legend for Ext Fig 3A--the dashed line seems to be the $y=x$ line, not the regression fit. If I am interpreting the graph correctly, then it appears that there is a systemic bias where the VAF is lower in blood than in the colonies for both B- and T-cells (dots are generally below the dotted line). I think this means that there are a substantial number of clones present in blood that are not able to be captured in the colonies, hence the lower VAF of the colony mutations in blood. This would be consistent with biased sampling in the colonies. The R^2 value presented is not so useful here -- if there is a consistent bias, the correlation will still look strong. If the authors agree that this is a valid interpretation of the data, I think it would be important to present this limitation as openly as possible in the text and discussion. If possible, also state what proportion of the T- and B-cell pool is "missing" in the in vitro expanded cells based on the expected versus observed VAFs for the mutations in blood.

The other points have been addressed to my satisfaction.

Referee #3 (Remarks to the Author):

Thank you for answering the questions, I have no further remarks.

Author Rebuttals to First Revision:

Referee #1 (Remarks to the Author):

The authors have done an outstanding job of responding to reviewer concerns and have further improved this elegant manuscript. The newly added analysis and characterization provides further insight into biology of B and T cells and will be of great interest to immunologists, while the added experimental detail will certainly also be appreciated.

In all, I find that the authors provided one of the most comprehensive responses to reviewers that I have seen and this manuscript will be a benchmark for studies into mutations and structural variations in hematopoietic system.

We thank the reviewer for this generous assessment of our manuscript.

Referee #2 (Remarks to the Author):

The authors now provide detail on the culturing methods and efficiencies, which shows that 0.5-14% are able to expand. This is considerably lower than their experiments on HSPCs, and could be a major source of confounding. To determine whether this leads to biased representation of cells in the sequence data, they perform three analyses.

First, they assess CDR3 regions of the sequenced cells. They do not find any redundant CDR3 sequences, suggesting that they are likely not expanding specific clones in vitro based on a TCR/BCR sequence. They also look at flow cytometric profiles of the sorted cells, which also confirms no evidence of bias by this metric. The final, and most important analysis, looks at the prevalence of specific mutations in colony expanded cells, versus in bulk cells from blood from one donor. This is the most important because it shows definitively whether the colony expanded cells are truly a random selection of all circulating cells based on proportional representation of clones in colonies vs blood. They plot the VAF of mutations in blood versus percent of colonies with the mutation in Ext Fig 3A. I believe there is an error in the figure legend for Ext Fig 3A--the dashed line seems to be the $y=x$ line, not the regression fit. If I am interpreting the graph correctly, then it appears that there is a systemic bias where the VAF is lower in blood than in the colonies for both B- and T-cells (dots are generally below the dotted line). I think this means that there are a substantial number of clones present in blood that are not able to be captured in the colonies, hence the lower VAF of the colony mutations in blood. This would be consistent with biased sampling in the colonies. The R2 value

presented is not so useful here -- if there is a consistent bias, the correlation will still look strong. If the authors agree that this is a valid interpretation of the data, I think it would be important to present this limitation as openly as possible in the text and discussion. If possible, also state what proportion of the T- and B-cell pool is "missing" in the *in vitro* expanded cells based on the expected versus observed VAFs for the mutations in blood.

We agree with the reviewer that this is an important issue to address fully, and that our analysis of the deep targeted sequencing data in the earlier revision of the manuscript was too superficial. The major question raised is whether the distribution of somatic mutations found in the colonies followed the distribution expected, given the frequency of those mutations in bulk lymphocyte populations. If the population of lymphocytes that successfully seeded colonies was not representative of all lymphocytes, then we would anticipate differences between the observed variant allele fractions (VAF) of mutations in bulk DNA and those seen across colonies – some lymphocyte lineages might grow more effectively *in vitro* than others, leading to over-representation of variants from those lineages in the colonies, and under-representation of variants from lineages that culture less efficiently.

To address this question more formally, we describe first the phylogenetic / lineage-tracing aspects of the detected mutations and second an analysis using non-parametric bootstrapping to assess whether there is evidence for bias in culture efficiency across lineages.

1. Phylogenetics and lineage relationships of somatic mutations

All somatic cells in an individual can trace their pedigree back through a series of cell divisions to the fertilised egg, which can be conceived as a vast lineage tree¹. Each of those cell divisions can generate new somatic mutations that become permanent lineage marks found in all subsequent descendants from that cell – the earliest branch-points in phylogenetic trees constructed this way represent cell divisions that predate even the split of placenta from the main foetus, for example³. The total number of lymphocytes in an adult human is in the order of trillions (10^{12}), whereas our deep sequencing for somatic mutations can detect mutations down to a frequency of ~ 0.003 with confidence. This means that it is only those mutations that occurred during embryonic and foetal development or mutations carried by monoclonal expansions (such as seen in clonal haematopoiesis) that will be detectable with our approach. In the individual studied here, we did not find evidence of sizable adult-onset clonal expansions in either haematopoietic stem cells or lymphocytes², so we would expect that the mutations detected through deep sequencing represent variants acquired during development.

In our earlier publication on this individual, we reconstructed the phylogenetic tree from his haematopoietic stem and progenitor cells, and showed that this tree included branches that

represented early embryonic cell divisions². We therefore mapped mutations called in the lymphocyte colonies and deep sequencing of B and T cells onto this phylogenetic tree – shown in **Reviewer Figure 1** below.

Reviewer Figure 1 (included as Supplementary Figure 1 in Supplementary Note). Variant allele fraction of mutations acquired during embryonic development in bulk and cultured lymphocyte populations from research subject AX001. A phylogenetic tree was reconstructed for AX001 from whole genome sequencing of haematopoietic stem and progenitor cells, as previously described² (top panel). The first 40 mutations of molecular time in this tree represent mutations acquired during foetal development³. Deep targeted sequencing was performed for those developmental mutations occurring in sufficiently unique genomic sequence for successful RNA bait design (about 47% of all mutations). The four panels below the overall tree show the same phylogenetic tree as the top panel, but zoomed in on the first 40 mutations of molecular time, with branches coloured by variant allele fraction from pink (low VAF/not targeted) to purple (high VAF) of those mutations in bulk DNA (left panels) or colonies (right panels) from B (upper) or T (lower) lymphocytes.

Several reassuring observations emerge from this comparison. First, the lymphocyte colonies draw broadly from across the developmental tree – the lymphocytes that we successfully cultured are truly polyclonal, deriving from both daughter cells of some of the earliest cell divisions in the embryo. Second, the fractional contributions of different developmental lineages to the lymphocyte colonies broadly matches that observed in bulk lymphocytes – the same lineages that contribute the highest fraction of cells to the bulk lymphocyte populations in the adult subject contribute similar fractions to the colonies (purple bars in **Reviewer Figure 1**). Third, comparing lineage VAFs between B and T lymphocytes reveals broadly similar distributions – these mutations occurred before the developmental split between B and T lymphocytes, and it is therefore reassuring that mutations are represented in approximately equal proportions across the two cell types.

2. Bootstrapping to quantify the extent of culture bias

As the reviewer states, the VAFs of mutations observed in bulk DNA versus in lymphocyte colonies showed spread around the expected equality line, with more points below that line than above in **Extended Figure 3A** – this could indicate systematic bias in culture efficiency among different lymphocyte lineages or, alternatively, could be entirely consistent with the expected effects of Poisson sampling. In statistical terms, the question is whether the *residuals* (that is, the deviation of points from the $y=x$ equality line) follow the distribution expected just from sampling noise, or whether their distribution is more consistent with effects from factors such as culture bias.

Standard statistical methods for assessing distributions of residuals (such as the Shapiro-Wilk test for normality) do not work in our scenario for the following two reasons:

- The counts of colonies reporting a mutation follow a Poisson distribution rather than

a normal distribution for a given mean – this leads to skewed residuals, especially when the expected VAF (μ) from bulk DNA is very low (the skewness of a Poisson distribution is $1/\sqrt{\mu}$ meaning that we would expect increasingly skewed residuals as the VAF gets closer to zero);

- The mutations are not independent of one another – this means that the VAF of mutations from the same lineage correlate with one another.

Therefore, to address the extent of any potential bias in the observed VAFs for colonies versus bulk DNA, we used nonparametric bootstrapping. Essentially, we draw samples of lymphocytes from the bulk lymphocyte trees shown in **Reviewer Figure 1** above. Each lineage is sampled at the frequency dictated by its VAF in the bulk lymphocyte population (including unobserved lineages, since these are evident as branch-points in the tree where the VAFs of descendant branches do not sum to the VAF of in the inbound branch). For any given lineage, we can introduce a simulated random bias into its culture efficiency by evolving this trait over the tree – for example, if the average culture efficiency is 0.05 then a culture bias of 20% would equate to efficiencies drawn from the range 0.04 to 0.06. For each bootstrap iteration, we sample simulated ‘lymphocytes’ from across the tree, maintaining those that are simulated to culture successfully using these lineage-specific ‘efficiencies’, until we acquire the correct number of ‘colonies’. Each iteration then generates a bootstrapped sample of colony VAFs to compare against the bulk lymphocyte VAFs. We generated 10,000 bootstrap iterations in this way for every level of culture bias from 0 to 0.9 in increments of 0.05.

We used QQ plots to assess our observed residuals against the expected distribution of residuals estimated from the bootstrap iterations. Where the level of culture efficiency bias is underestimated, the observed residual quantiles will be above the equality line in the QQ plots (because the residuals are systematically larger than expected); where culture bias is overestimated, the residual quantiles will fall below the equality line. The expected ranges of residuals and the QQ plots are shown in **Reviewer Figure 2**.

Reviewer Figure 2 (included as Supplementary Figure 2). Distributions of observed versus expected residuals for colony VAFs under different levels of bias in culture efficiency. The four rows represent different levels of bias in lineage-specific culture efficiency in the nonparametric bootstrap samples (0%, 20%, 40% and 60%), with bootstraps for B lymphocytes on the left and T lymphocytes on the right. Within each condition, the left graph shows the observed data for VAFs in bulk DNA (x axis) versus colonies (y axis) as black points distributed around the x=y line in grey. The shaded areas represent the estimated 90% (blue), 95% (orange) and 99% (green) intervals for the distributions of residuals from the bootstrap samples. The right graph shows the QQ plot with the expected (bootstrap) quantiles on the x axis and the observed quantiles on the y axis. The x=y equality line is shown as a dashed black line.

Reassuringly, we see a strong correlation between the bulk and colony VAFs – the QQ plots for the condition of zero culture bias are close to the equality line (top row of plots in **Reviewer Figure 2**) indicating that the observed residuals are distributed similar to expectation. Nonetheless, for both B and T cells, there is a small systematic deviation above the equality line at zero culture bias; as the level of bias in culture efficiency increases, the QQ plots initially move closer to the equality line and then beyond. If we express this using an ‘area under the curve’ (AUC; specifically the area above or

below the equality line), we see that this area is closest to 0 when the bias in culture efficiency is ~20% for B and T cells (**Reviewer Figure 3**). Thus, if the discrepancy between colony and bulk VAFs were entirely explained by lineage-specific bias in culture efficiency, we estimate that it would equate to a bias of ~20% for both B and T cells (culture probabilities typically within the range 0.04 to 0.06 for a given lineage).

We have included the analyses above as a Supplementary Note in the manuscript, quoted the estimates of potential bias in the main text ('Genome sequencing of B and T lymphocytes'; p. 3), provided details of how the bootstrapping analysis was undertaken in the Methods and provided the code for the analysis in the github repository for the paper.

Reviewer Figure 3 (included as Supplementary Figure 3). Area under the curve (AUC) for QQ plots for bootstraps performed with different levels of bias in culture efficiency. The AUC is measured as the net area above or below the equality line in the QQ plots as shown in Reviewer Figure 2. Positive values for AUC suggest that the bias is systematically underestimated, whereas negative values suggest it is overestimated. The lines cross 0 when the level of bias in culture efficiency is ~20% for both T and B lymphocytes.

The other points have been addressed to my satisfaction.

We are pleased that the other points have been appropriately addressed, and hope that the further clarification above on the question of culture efficiency is satisfactory.

Referee #3 (Remarks to the Author):

Thank you for answering the questions, I have no further remarks.

We thank the referee for the careful review of our manuscript.

References

1. Stratton, M. R., Campbell, P. J. & Futreal, P. A. The cancer genome. *Nature* **458**, 719–724 (2009).
2. Lee-Six, H. *et al.* Population dynamics of normal human blood inferred from somatic mutations. *Nature* **561**, 473–478 (2018).
3. Spencer Chapman, M. *et al.* Lineage tracing of human development through somatic mutations. *Nature* **595**, 85–90 (2021).

Reviewer Reports on the Second Revision:

Referees' comments:

Referee #2 (Remarks to the Author):

The authors have addressed my concern about a possible bias in cells which grow out in culture. In both the Ext Fig 3 figure legend and supplementary note they still describe a "linear regression fit" line. This appears to be the equality line instead of an OLS fit. If so, they may wish to correct this to avoid confusing the reader.

Author Rebuttals to Second Revision:

Referees' comments:

Referee #2 (Remarks to the Author):

The authors have addressed my concern about a possible bias in cells which grow out in culture. In both the Ext Fig 3 figure legend and supplementary note they still describe a "linear regression fit" line. This appears to be the equality line instead of an OLS fit. If so, they may wish to correct this to avoid confusing the reader.

We now show both the equality line (dashed line) and the line fit by linear regression (solid line) in Extended Figure 2A (previously Extended Figure 3A).